# TOWARDS EFFICIENT ADAPTATION OF PRUNING STRATEGY IN LARGE LANGUAGE MODELS

## ABSTRACT

Post-training pruning has gained increasing attention with the rapid growth of large language models (LLMs). However, significant variations in weight distributions across different LLMs make a fixed pruning strategy inadequate for multiple models. In this paper, we propose an efficient evolutionary optimization framework, **MECON**, for adaptive LLM pruning. In particular, we design an effective search space built on our **Me**ta pruning metric to mitigate diverse weight distributions among LLMs. We then introduce model-wise re**con**struction error, a lightweight search evaluation to speed up the evaluation of each search trial. We finally leverage Non-dominated Sorting Genetic Algorithm III (NSGA-III) as our search algorithm, handling both the single-objective problem of pruning metric search and the multi-objective problem of layerwise sparsity ratio search in discovering the optimal pruning strategy. We extensively evaluate our framework on LLaMA-1/2/3 and Mistral models across multiple benchmarks. Our results demonstrate that our adaptive pruning metrics consistently outperform existing ones, and the layerwise sparsity ratios improve the effectiveness of other pruning metrics. Furthermore, we validate the cross-task and cross-model generalizability of our pruning metrics, offering a cost-effective solution to streamline the search process. We release our code in the anonymous repository: https://anonymous.4open.science/r/Mecon-5819.

## 1 INTRODUCTION

Large language models (LLMs) (Achiam et al., 2023; Touvron et al., 2023a; Le Scao et al., 2023) have recently shown remarkable performance in a range of complex language benchmarks in the field of language understanding and generation (Bubeck et al., 2023; Wei et al., 2022a;b). Despite their impressive performance, their extensive model size causes significant computational demands, making LLM inference and deployment a big challenge. One notable advancement in model compression has centered on model pruning (LeCun et al., 1989; Hassibi et al., 1993; Han et al., 2015), which shrinks model sizes by removing specific weights from the model – essentially setting them to zero. Traditional model pruning methods, typically involve retraining (Liu et al., 2018; Blalock et al., 2020) or iterative training to recover performance (Frankle & Carbin, 2018; Renda et al., 2019), which are less feasible when scaling to large LLMs with billions of parameters. Recently, there has been a growing effort in post-training pruning (PTP) due to its minimal resource demands. PTP methods develop pruning metrics to evaluate the importance of weights, thus the weights with lower importance can be removed. (Frantar & Alistarh, 2023; Sun et al., 2023; Zhang et al.).

However, as shown in Figure 1, we observe a significant performance drop when applying recent well-established pruning metrics (Frantar & Alistarh, 2023; Sun et al., 2023; Zhang et al.) to the LLaMA-3 (Meta, 2024) model. To analyze the reason for the performance drop, we demonstrate the distributions of input activation norms and weight magnitudes, two main components considered by recent pruning metrics. Despite the past success of SparseGPT (Frantar & Alistarh, 2023), Wanda (Sun et al., 2023), and RIA (Zhang et al.) on LLaMA-1 (Touvron et al., 2023a) and LLaMA-2 (Touvron et al., 2023b) models, the distinct weight distribution of LLaMA-3 (Meta, 2024) underscores the limitations of using a fixed pruning metric across LLMs with varying weight distributions.

In this paper, we study the essential adaption of pruning strategy across different LLMs, and propose an efficient evolutionary optimization framework, named MECON, to automatically search for the

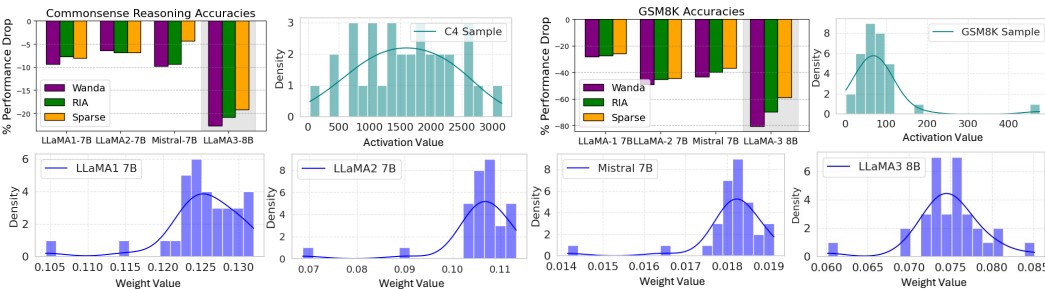

Figure 1: **Performance of existing pruning metrics on different LLMs.** Existing pruning metrics show significant performance drops on the LLaMA-3 model (bar charts in the upper part), influenced by its distinct weight distribution compared to LLaMA-1/2 and Mistral models (the lower part).

adaptive pruning strategy for different LLMs, including optimization of both the pruning metric and the layerwise sparsity ratios. In particular, we design an effective search space built on our Meta pruning metric. Unlike prior pruning metrics (Frantar & Alistarh, 2023; Sun et al., 2023; Zhang et al.) that consider weights and activations rely on fixed heuristics, our meta pruning metric dynamically balances the relationship between weights and activations, to mitigate diverse weight distributions among different LLMs. Moreover, we also consider a better way for post-training pruning evaluations of each search trial. We show that prior evaluations based on perplexity (Dong et al., 2024) are more time-consuming and establish limited generalizability across different downstream tasks. Instead, we propose a lightweight search evaluation, model-wise reconstruction error, to speed up the evaluation in each search trial. Finally, we apply Non-dominated Sorting Genetic Algorithm III (NSGA-III) (Deb & Jain, 2013; Jain & Deb, 2013) as our search algorithm, handling both the single-objective problem of pruning metric search and the multi-objective problem of layerwise sparsity ratio search in a unified framework.

We empirically evaluate MECON on the widely adopted LLaMA-1 (Touvron et al., 2023a), LLaMA-2 (Touvron et al., 2023b), LLaMA-3 (Meta, 2024) and Mistral (Jiang et al., 2023) models across multiple benchmarks. Our results demonstrate that, without any retraining or weight update, our MECON-derived pruning metrics consistently outperform all established pruning metrics. Additionally, our MECON-derived layerwise sparsity ratios could also boost the effectiveness of other pruning metrics that consider both weight and activation, such as Wanda (Sun et al., 2023) and RIA (Zhang et al.). Furthermore, we verify the generalizability of our MECON-derived pruning metrics through cross-task and cross-model evaluations, showing that metrics developed for complex arithmetic reasoning tasks also perform well on simpler tasks like commonsense reasoning and language modeling, and remain effective when applied to models of different configurations. Thus we provide a cost-effective alternative to streamline the adaptive search process.

## 2 RELATED WORK

**Emergent Large Features of LLMs** Emergent large magnitude and massive activation features have been observed in Transformer-based large language models (Kovaleva et al., 2021; Puccetti et al., 2022; Wei et al., 2022c; Dettmers et al., 2022; Sun et al., 2024). The occurrence of these hidden state features and input activations with large magnitudes is relatively rare, indicating the outlier patterns within model internal representations. However, these outlier features are shown to have essential importance in representing information, as zeroing out these outlier features during inference leads to a significant degradation in model performance (Dettmers et al., 2022; Sun et al., 2024). Recent development of quantization schemes (Lin et al., 2023; Dettmers et al., 2023; Xiao et al., 2023) and model pruning methods (Sun et al., 2023; Zhang et al.) for LLMs have been influenced by the presence of these outlier features. Our research expands on this insight by demonstrating that the relationship between these weights and input activation outlier features should also act as key indicators for selecting which weights to prune in LLMs.

**Post-Training Pruning** Post training pruning (PTP) has emerged as a popular technique for reducing the size and computational complexity of models without the need for extensive retraining (Hubara et al., 2021; Kwon et al., 2022; Frantar & Alistarh, 2023). Recent PTP methods for LLMs aim to

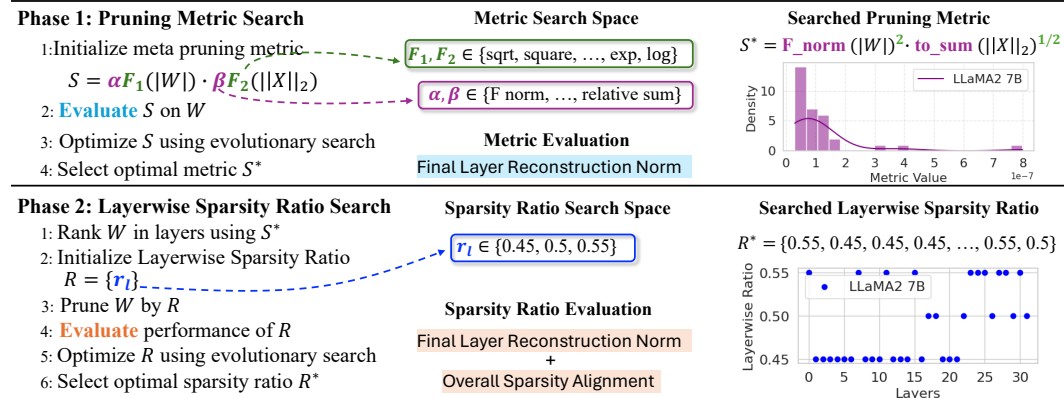

Figure 2: **Illustration of our proposed MECON method.** MECON consists of two phases: searching for the optimal pruning metric and the optimal layerwise sparsity ratios. We present the distribution of metric scores for the optimal metric searched in the LLaMA2-7B model, along with the sparsity ratio for each layer in the right column.

evaluate the importance of weights using specific pruning metrics and remove less important weights by setting them to zero. Magnitude pruning (Han et al., 2015) directly removes weights based on their magnitude, offering simplicity but often resulting in unsatisfied performance for LLMs. To improve accuracy, SparseGPT (Frantar & Alistarh, 2023) solves layer-wise reconstruction problem, which significantly boosts performance but adds computational costs due to weight updates. Wanda (Sun et al., 2023) simplifies SparseGPT by considering only the product of weight magnitude and the norm of input activations. Building on Wanda, RIA (Zhang et al.) introduces a relative importance coefficient to enhance weight importance evaluation. These one-shot pruning metrics now stand out as strong baseline approaches for LLM pruning.

## 3 MECON: ADAPTIVE PRUNING STRATEGY SEARCH

### 3.1 METHOD OVERVIEW

MECON focuses on the essential adaption of pruning strategy across different LLMs. As shown in Figure2, MECON automatically searches for the adaptive pruning strategy through evolutionary optimization, optimizing both the pruning metric and layerwise sparsity ratios. In each phase of the search, MECON iteratively samples pruning metrics or layerwise sparsity ratios from a predefined search space. Each sample is then evaluated, producing a numerical measure of its quality. The evaluation results are fed back into the search algorithm to improve future samplings. The details of these two phases are as follows:

- **Pruning Metric Search** - involves identifying the most effective metric for scoring the importance of model weights. Similar to Wanda (Sun et al., 2023), we compare the weights on a per-output basis, where weight importance is assessed locally within each output neuron.

- **Layerwise Sparsity Ratio Search** - determines the optimal non-uniform sparsity ratios for different layers in the model. After assigning importance scores to the weights using the pruning metric, we prune the weights with lower scores according to the specified sparsity ratio for each layer.

In Section 3.2, we outline the Search Space for each phase of MECON, covering the range of pruning metrics and sparsity ratios that can be explored. Section 3.3 describes the Evaluation Measurement, detailing how we efficiently assess the sampled pruning metrics and layerwise sparsity ratios. Lastly, in Section 3.4, we describe the search algorithm, highlighting how we leverage an evolutionary approach to identify the optimal pruning metric and layerwise sparsity ratios.

### 3.2 SEARCH SPACE

**Meta Pruning Metric.** Inspired by the discovery of emergent large weight magnitude (Puccetti et al., 2022; Wei et al., 2022c; Dettmers et al., 2022) and massive input activation (Sun et al., 2024) features in LLMs, recent pruning metrics (Sun et al., 2023; Zhang et al.) find that augmenting the

standard weight magnitude pruning metric with the input activations shows great effectiveness in evaluating the weight importance.

Building upon these concepts, we introduce a meta pruning metric to dynamically balance the relationship between weights and activations for LLMs with varying weight distributions. As shown in Equation 1, we score each weight $W_{ij}$ by applying a weighted transformation to the element-wise product between the weight magnitude $|W_{ij}|$ and the norm of input activations $\|X_j\|_2$. The weighted transformation is conducted by specific coefficients ($\alpha$, $\beta$) and operations ($F_1$, $F_2$).

$$S_{ij} = \alpha F_1(|W_{ij}|) \cdot \beta F_2(\|X_j\|_2). \tag{1}$$

Here, we use the absolute value of weights $|W_{ij}|$ to calculate weight magnitudes and $\|X_j\|_2$ to measure the norm of input activations, which computes the $l_2$ norm of the $j_{th}$ features aggregated across different tokens. The predefined coefficients and operation candidates are listed in Table 1, with further details and calculation equations provided in Appendix A.5.

Notably, our meta pruning metric is able to encompasses and extends beyond existing pruning metrics like Wanda and RIA. For instance, the Wanda metric (Sun et al., 2023), expressed as

$$S_{ij} = |W_{ij}| \cdot \|X_j\|_2, \tag{2}$$

does not set coefficients and operations, providing a uniform weighting between weight magnitude and norm of input activations. Building on this, the RIA metric (Zhang et al.), denoted as

$$S_{ij} = \mathrm{RI}(|\mathrm{W_{ij}}|) \cdot \|\mathrm{X_j}\|_2^{1/2}, \tag{3}$$

modifies the coefficient of the weight magnitude as relative sum $\mathrm{RI}$, and sets the operation of the input activation norm as a square function.

Table 1: Predefined coefficient and operation candidates for Meta Pruning Metric.

| coefficient candidates for $\alpha, \beta$ | no coe, F norm, to sum, to mean, row sum, column sum, relative sum |
|---|---|
| operation candidates for $F_1, F_2$ | no op, sqrt, square, sigmoid, softmax, exp, log |

**Layerwise Sparsity Ratios.** Within Transformer architectures, neurons across different layers are observed to capture distinct types of information (Wang & Tu, 2020; Zhang et al., 2021), thus exhibiting different priorities in maintaining original performance. To leverage this insight, we prune LLMs in a non-uniform layerwise sparsity, for layers with more important neurons, we set a lower pruning ratio, while layers with less important neurons are assigned a higher pruning ratio. Specifically, we identify the optimal sparsity ratio for each layer by selecting from a predefined sparsity ratio set, including `target sparsity - sparsity step`, `target sparsity` and `target sparsity + sparsity step`. Here, target sparsity is the pre-defined sparsity ratio for pruning the overall model. The sparsity step allows for adjustments to achieve slightly higher or lower sparsity ratios, facilitating non-uniform sparsity across different layers. We empirically find that for LLMs with more than 32 layers, using a discrete set of three sparsity ratios outperforms larger sets when searching within a limited number of trials.

### 3.3 Search Evaluation

Search Evaluation measures each sampling from the search space, thus guiding the evolutionary process toward finding the optimal pruning strategy (Bäck & Schwefel, 1993). The primary goal of model pruning is to remove a subset of network weights while aiming to preserve performance (LeCun et al., 1989; Han et al., 2015). Following this goal, MECON leverages model-wise reconstruction error to evaluate each sampled pruning metric. Furthermore, we introduce a secondary measurement to assess the overall sparsity ratio of the pruned model to evaluate sampled layerwise sparsity ratios.

**Model-wise Reconstruction Error.** We propose a lightweight search evaluation, model-wise reconstruction error, for the sampled pruning strategy. Existing automatic framework, like Pruner-Zero (Dong et al., 2024), use perplexity as the evaluation measure. However, we demonstrate that using perplexity requires more evaluation time and tends to generalize poorly across different downstream tasks. Toward that end, the proposed model-wise reconstruction error admits a faster evaluation of each search trial while preserving generalizability.

Mathematically, the model-wise reconstruction error, denoted as $f_{rec}$, measures the discrepancy norm in the final layer outputs between the dense model $\theta$ and the pruned model $\theta^*$. Specifically, we denote the input activations for the final layer $l$ as $X_l$, and weight with $r$ output channels and $c$ input channels as $W_l \in \mathbb{R}^{r \times c}$. A layerwise sparsity mask $M_i \in {0,1}^{r \times c}, i \in \{0, ..., l\}$ removes a certain degree of model weights with lower importance scores, which are measured by the sampled pruning metric. Therefore, the model-wise reconstruction error can be formally expressed as:

$$f_{rec}(\theta, \theta^*) = \|W_l X_l - (M_l \odot W_l) \cdot X_l\|_{Frob}, \tag{4}$$

where $||\cdot||_{Frob}$ is the Frobenius norm (Golub & Van Loan, 1996), ensuring the final output of pruned model $\theta^*$ closely matches that of dense model $\theta$.

**Sparsity Ratio Discrepancy.** Layerwise sparsity search assigns each layer a sampled sparsity ratio which is slightly higher or lower than the pre-defined sparsity ratio. As a result, the overall sparsity of the pruned model may deviate from the pre-defined ratio to prune the dense model. Thus, we introduce a secondary measurement, sparsity ratio discrepancy, to evaluate the numerical difference between the sparsity ratio of the pruned model and the pre-defined sparsity ratio. The sparsity ratio discrepancy $f_{ratio}$ is mathematically defined as:

$$f_{ratio}(\theta, \theta^*) = |R_d - \frac{\text{parameters}(\theta) - \text{parameters}(\theta^*|\mathcal{R})}{\text{parameters}(\theta)}|. \tag{5}$$

Here, $R_d$ denotes the pre-defined sparsity ratio for pruning the dense model, and $\mathcal{R}$ is the layerwise sparsity ratios applied to the pruned model $\theta^*$. The function $\text{parameters}(\cdot)$ measures the total number of parameters in the model. Therefore, the sparsity ratio of the pruned model is thus calculated by comparing the number of removed parameters to the total number of parameters in the dense model.

### 3.4 Search Algorithm

We employ the Non-dominated Sorting Genetic Algorithm III (NSGA-III) Deb & Jain (2013) as our search algorithm. NSGA-III is capable of handling both single and multi-objective optimization problems, making it suitable for addressing both pruning metric search and layerwise sparsity ratio search scenarios within a unified framework. Let $\mathcal{S} = \{s_1, s_2, ..., s_n\}$ be the search space, where $s_i$ represents a candidate pruning strategy. We define the objective function as $f(s) : \mathcal{S} \rightarrow \mathbb{R}$, where $f(s)$ is to be minimized. For the single-objective problem of pruning metric search, we aim to find:

$$s^* = \underset{s \in \mathcal{S}}{\arg\min} f(s) \tag{6}$$

---

**Algorithm 1** NSGA-III for Searching Adaptive Pruning Strategy

---

1: Initialize population $P_0$ of size $N$
2: **for** $t = 1$ to $T$ **do**
3:     $Q_t \leftarrow$ CreateOffspring($P_t$)
4:     $R_t \leftarrow P_t \cup Q_t$
5:     $\mathcal{F} \leftarrow$ NonDominatedSort($R_t$)
6:     $P_{t+1} \leftarrow$ SelectPopulation($\mathcal{F}, N$)
7: **end for**
8: **return** $P_T$

---

For the multi-objective problem of layerwise sparsity ratio search, we define a vector of objective functions $\mathbf{F}(s) = (f_1(s), f_2(s), ..., f_k(s))$. The goal is to find the Pareto optimal set:

$$\mathcal{S}^* = \{s \in \mathcal{S} \mid \nexists s' \in \mathcal{S} : \mathbf{F}(s') \prec \mathbf{F}(s)\} \tag{7}$$

where $\prec$ denotes Pareto dominance. The dominance relation is defined as: A solution $x_1$ dominates $x_2$ ($x_1 \prec x_2$) if and only if:

$$\forall i \in \{1, ..., k\} : f_i(x_1) \leq f_i(x_2) \wedge \exists j \in \{1, ..., k\} : f_j(x_1) < f_j(x_2) \tag{8}$$

Specifically, MECON follows a two-phase search process. In the first phase, we minimize the model-wise reconstruction error $f_{\text{rec}}$ (Eq. 4) to find the optimal pruning metrics. During the next phase, we aim to find the optimal layerwise sparsity ratios by minimizing both $f_{\text{rec}}$ and the sparsity ratio discrepancy $f_{\text{ratio}}$ (Eq. 5).

The detailed process of NSGA-III for pruning strategy search is provided in Algorithm 1. The algorithm starts with an initial population $P_0$ and iterates for a fixed number of generations. In each generation, it combines the parent $P_t$ and offspring populations $Q_t$, performs non-dominated sorting to rank solutions, and selects the best solutions to form the next generation $P_{t+1}$. The niching procedure ensures diversity by favoring solutions close to under-represented reference points.

Table 2: Mean zero-shot accuracies(%) on the LM Harness and WikiText perplexity of pruned LLaMA-1/2/3 and Mistral models.

| Method | Weight Update | Sparsity | LLaMA-1 | | LLaMA-2 | | LLaMA-3 | | Mistral | |
|---|---|---|---|---|---|---|---|---|---|---|
| | | | 7B | 13B | 7B | 13B | 8B | 8B-Inst | 7B | 7B-Inst |
| LM Harness | | | | | | | | | | |
| Dense | - | 0% | 59.70 | 62.58 | 59.72 | 63.03 | 64.21 | 64.15 | 60.06 | 66.69 |
| Magnitude | ✗ | 50% | 46.89 | 47.34 | 52.40 | 52.90 | 44.87 | 45.31 | 57.24 | 63.34 |
| SparseGPT | ✓ | 50% | 54.86 | 58.54 | 55.90 | 60.70 | 53.87 | 55.89 | 57.49 | 62.46 |
| Wanda | ✗ | 50% | 54.08 | 59.18 | 55.89 | 60.88 | 49.66 | 51.34 | 54.20 | 61.04 |
| RIA | ✗ | 50% | **55.10** | 59.45 | 55.67 | 61.03 | 50.76 | 50.64 | 54.39 | 60.48 |
| Pruner-Zero | ✗ | 50% | 52.31 | 57.08 | 53.81 | 58.18 | 52.48 | 55.60 | 55.57 | 61.41 |
| MECON | ✗ | 50% | **55.10** | **59.73** | **57.47** | **61.42** | **55.50** | **55.94** | **59.33** | **63.51** |
| WikiText Perplexity | | | | | | | | | | |
| Dense | - | 0% | 5.37 | 4.80 | 5.04 | 4.56 | 5.80 | 7.91 | 5.23 | 4.90 |
| Magnitude | ✗ | 50% | 13.27 | 13.55 | 11.96 | 6.16 | 73.93 | 5.5E2 | 7.14 | 6.59 |
| SparseGPT | ✓ | 50% | 6.92 | 5.87 | 6.59 | 5.72 | 10.89 | 13.27 | 6.42 | 7.02 |
| Wanda | ✗ | 50% | 6.90 | 5.82 | 6.47 | 5.64 | 10.57 | 16.37 | 7.24 | 7.22 |
| RIA | ✗ | 50% | 6.81 | 5.83 | 6.43 | 5.63 | 12.56 | 15.57 | 7.27 | 7.21 |
| Pruner-Zero | ✗ | 50% | 7.13 | 6.02 | 6.86 | 5.88 | 12.68 | 15.45 | 7.84 | 7.50 |
| MECON | ✗ | 50% | **6.78** | **5.74** | **6.35** | **5.51** | **9.23** | **11.37** | **6.22** | **6.55** |

## 4 EXPERIMENTS

### 4.1 SETUP

**Models and Evaluations.** To demonstrate the effectiveness of MECON, we adopt four prominent open-sourcing large language models as our foundation model, including LLaMA-1 (Touvron et al., 2023a) and LLaMA-2 (Touvron et al., 2023b) with sizes ranging from 7B to 70B, LLaMA-3 8B (Meta, 2024) and Mistral 7B (Jiang et al., 2023) with the base models and their instruction-tuned variants. Following previous works (Sun et al., 2023; Xia et al., 2023), we first evaluate on seven tasks from the EleutherAI LM Harness (Gao et al., 2023)[1], and the language modeling task based on the held-out WikiText (Merity et al., 2016) validation set. Furthermore, we also evaluate two more challenging tasks, namely arithmetic reasoning on GSM8K (Cobbe et al., 2021) and the language understanding benchmark MMLU (Hendrycks et al., 2020). For the comparison group settings, we follow Wanda (Sun et al., 2023) to compare and remove weights on a per-output basis, where weight importance scores are compared locally within each output neuron. We evaluate three sparsity types as defined in previous research (Sun et al., 2023; Zhang et al.): unstructured sparsity, semi-structured 4:8 and 2:4 sparsity. We set the number of trails in the search process as 350, further details on hyperparameter analysis are provided in Appendix A.4.

**Baselines.** We compare MECON with five existing outstanding baselines. Magnitude pruning (Han et al., 2015) is a straightforward but effective solution which discards weights based on their magnitudes. SparseGPT (Frantar & Alistarh, 2023) solves the layer-wise reconstruction problem to identify redundant weights and prune them accordingly. Wanda (Sun et al., 2023) utilizes large-magnitude features and input activation to induce sparsity. RIA (Zhang et al.) further improves Wanda pruning by introducing the relative importance and channel permutation. Pruner-Zero (Dong et al., 2024) automatically searches for the optimal pruning metric based on weights and gradients, using perplexity on WikiText as the evaluation measure.

**Calibration Data.** Calibration data is used to estimate input statistics from a small set of samples. For a fair comparison, we use the exact same calibration data as Wanda and SparseGPT when evaluating on LM Harness and WikiText, which includes 128 sequences sampled from the C4 training set (Raffel et al., 2020). For evaluations on GSM8K and MMLU, we randomly select 10 samples from the training dataset, each truncated to a sequence length of 512, as our calibration samples.

### 4.2 MAIN RESULTS

**LM Harness & Language Modeling** Table 2 presents the performance of LM Harness and the WikiText perplexity on the language modeling task. We refer the reader to Appendix A.7 for task-wise

---

[1]Referred as *LM Harness* in remaining parts.

Table 3: WikiText perplexity and mean zero-shot accuracies(%) on the LM Harness of 50% unstructured pruned LLaMA-1 30B and LLaMA-2 70B models.

| Model | Method | WikiText | BoolQ | RTE | HellaSwag | WinoGrande | ARC-e | ARC-c | OBQA | Average |
|---|---|---|---|---|---|---|---|---|---|---|
| LLaMA 30B | Dense | 4.77 | 82.69 | 66.79 | 63.35 | 75.69 | 80.30 | 52.82 | 36.00 | 65.38 |
| | Magnitude | 7.55 | 64.34 | 50.18 | 50.59 | 66.54 | 72.39 | 43.77 | 29.00 | 53.83 |
| | SparseGPT | 5.32 | 82.32 | 62.45 | 59.15 | 75.22 | 78.96 | 48.56 | **35.00** | 63.09 |
| | Wanda | 5.98 | 81.90 | 65.34 | **60.93** | **73.48** | **79.29** | 49.66 | 34.60 | 63.60 |
| | RIA | 5.16 | **83.36** | 67.15 | 60.01 | 72.85 | 78.70 | 48.29 | 33.60 | 63.42 |
| | MECON | **5.10** | **83.36** | **67.51** | **60.93** | 72.61 | 78.91 | **49.74** | 34.20 | **63.89** |
| LLaMA2 70B | Dense | 3.12 | 83.40 | 67.87 | 66.10 | 78.06 | 82.55 | 54.44 | 37.20 | 67.08 |
| | Magnitude | 4.98 | 70.55 | 60.65 | 61.50 | 73.48 | 75.70 | 49.23 | 35.40 | 60.93 |
| | SparseGPT | 3.98 | **83.55** | 70.40 | 63.80 | **78.85** | **82.40** | 53.75 | 38.20 | 67.28 |
| | Wanda | 3.99 | 82.50 | **73.65** | **64.10** | 78.14 | 80.80 | 52.65 | 37.40 | 67.03 |
| | RIA | 3.91 | 83.25 | 71.49 | 64.05 | 77.74 | 81.20 | 53.16 | 36.60 | 66.77 |
| | MECON | **3.86** | 83.25 | 73.21 | 64.00 | 78.48 | 81.25 | 53.07 | **38.40** | **67.38** |

performance. The results indicate that our method consistently outperforms all established baselines across the board. More intriguingly, a notable performance gap between SparseGPT and the other two baselines, i.e. Wanda and RIA, is observed on the LLaMA-3 and Mistral models, while the performance remains comparable on the LLaMA-1 and LLaMA-2 models. This observation is also aligned with the results in previous work (Sun et al., 2023). However, our MECON framework further significantly improves the state-of-the-art performance on all types of models. We think this is attributed to the different weight distributions of LLaMA-1/2 models with the LLaMA-3 model, as depicted in Figure 1, highlighting the importance of adaptive pruning on different models.

We also explore the effectiveness of MECON when appied to larger models, such as LLaMA-30B and LLaMA2-70B. As shown in Table 3, MECON consistently achieves the best performance on both the WikiText perplexity and LM Harness benchmarks. Besides, for the particularly larger models like LLaMA2-70B, MECON can even outperform the dense model without the need for weight updating.

Table 4: GSM8K and MMLU accuracies(%) of pruned LLaMA-1/2/3 and Mistral models.

| Method | Weight Update | Sparsity | LLaMA-1 | | LLaMA-2 | | LLaMA-3 | | Mistral | |
|---|---|---|---|---|---|---|---|---|---|---|
| | | | 7B | 13B | 7B | 13B | 8B | 8B-Inst | 7B | 7B-Inst |
| GSM8K | | | | | | | | | | |
| Dense | - | 0% | 11.07 | 17.82 | 14.59 | 19.86 | 52.39 | 74.45 | 40.11 | 47.76 |
| Magnitude | ✗ | 50% | 1.52 | 5.99 | 2.05 | 6.22 | 1.97 | 1.29 | 15.53 | 27.37 |
| SparseGPT | ✓ | 50% | 8.19 | 15.60 | 8.11 | 13.42 | 21.46 | 49.20 | 25.40 | 33.97 |
| Wanda | ✗ | 50% | 7.96 | 11.52 | 7.43 | 9.10 | 10.16 | 32.68 | 22.74 | 33.59 |
| RIA | ✗ | 50% | 8.04 | 11.14 | 7.96 | 9.25 | 15.85 | 52.39 | 24.18 | 32.15 |
| Pruner-Zero | ✗ | 50% | 6.41 | 9.22 | 7.32 | 8.58 | 17.25 | 43.63 | 21.16 | 32.24 |
| MECON | ✗ | 50% | 8.14 | 15.37 | 8.13 | 13.79 | 41.17 | **52.39** | 25.31 | **35.25** |
| w/ eval. | ✗ | 50% | **8.22** | **15.62** | **8.47** | **15.03** | **43.07** | 52.15 | **25.78** | 35.14 |
| MMLU | | | | | | | | | | |
| Dense | - | 0% | 35.28 | 46.98 | 41.97 | 51.47 | 65.23 | 66.35 | 58.92 | 62.54 |
| Magnitude | ✗ | 50% | 26.24 | 30.12 | 26.04 | 43.83 | 4.36 | 12.03 | 50.83 | 49.52 |
| SparseGPT | ✓ | 50% | 29.48 | 38.29 | 33.03 | 47.14 | 49.50 | 52.27 | 50.95 | 52.04 |
| Wanda | ✗ | 50% | 29.81 | 37.84 | 32.09 | 48.06 | 49.05 | 53.15 | 53.05 | 53.62 |
| RIA | ✗ | 50% | 30.37 | 37.79 | 31.46 | 47.39 | 48.99 | 54.02 | 52.67 | 53.14 |
| Pruner-Zero | ✗ | 50% | 28.57 | 35.51 | 30.26 | 45.24 | 41.39 | 46.32 | 51.75 | 53.15 |
| MECON | ✗ | 50% | 30.93 | 38.80 | 32.24 | 48.15 | 50.65 | 55.11 | 53.10 | 53.77 |
| w/ eval. | ✗ | 50% | **31.05** | **39.76** | **33.06** | **48.38** | **51.22** | **55.60** | **53.87** | **54.36** |

**Arithmetic & Knowledge Reasoning** In Table 4, we report the performance of pruned LLaMA-1/2/3 and Mistral models on the GSM8K and MMLU dataset. We can see that MECON consistently outperforms all baselines on the reasoning tasks. We highlight that we make remarkable improvements on the GSM8K dataset. For instance, on the LLaMA-3 8B model, MECON achieves an accuracy of 41.17, significantly better than the previous best performance of 21.46. This result also suggests that existing pruning methods are sensitive to the models. Additionally, since the optimal pruning strategies differ across the tasks but MECON is task-agnostic, we also attempt to align the search process of MECON with the target task objective. We implement it by introducing the evaluation accuracy on the validation set as an additional search objective. We find that with the aid of evaluation accuracy, further improvements are achieved over the standard MECON.

Table 5: Evaluations of semi-structured N:M sparsity on WikiText and GSM8K datasets.

| Method | Weight Update | Sparsity | LLaMA-1 | | LLaMA-2 | | LLaMA-3 | | Mistral | |
|---|---|---|---|---|---|---|---|---|---|---|
| | | | 7B | 13B | 7B | 13B | 8B | 8B-Inst | 7B | 7B-Inst |
| WiKiText Perplexity | | | | | | | | | | |
| Magnitude | ✗ | 4:8 | 17.48 | 16.80 | 16.10 | 7.23 | 2.5E2 | 5.6E2 | 8.78 | 8.67 |
| SparseGPT | ✓ | 4:8 | 8.16 | 7.05 | 7.89 | 6.54 | **15.57** | **16.62** | 7.71 | 8.15 |
| Wanda | ✗ | 4:8 | 8.19 | 6.95 | 8.01 | 6.60 | 16.82 | 21.52 | 8.95 | 8.42 |
| RIA | ✗ | 4:8 | 8.18 | 6.97 | 8.04 | 6.62 | 17.28 | 21.15 | 8.91 | 8.51 |
| MECON | ✗ | 4:8 | **7.93** | **6.65** | **7.72** | **6.34** | 17.24 | 21.15 | **7.57** | **7.66** |
| Magnitude | ✗ | 2:4 | 49.06 | 19.33 | 38.50 | 9.04 | 5.3E3 | 5.3E3 | 13.18 | 11.83 |
| SparseGPT | ✓ | 2:4 | 10.58 | 8.53 | 10.38 | 8.26 | **23.43** | **26.68** | 10.17 | 9.84 |
| Wanda | ✗ | 2:4 | 11.04 | 9.06 | 11.31 | 8.46 | 31.89 | 59.12 | 13.54 | 11.08 |
| RIA | ✗ | 2:4 | 11.10 | 9.24 | 11.40 | 8.57 | 31.79 | 38.00 | 13.61 | 11.21 |
| MECON | ✗ | 2:4 | **10.54** | **8.21** | **10.34** | **7.97** | 31.71 | 37.98 | **10.13** | **9.23** |
| GSM8K | | | | | | | | | | |
| Magnitude | ✗ | 4:8 | 1.53 | 3.48 | 1.59 | 4.70 | 4.16 | 7.81 | 9.60 | 14.15 |
| SparseGPT | ✓ | 4:8 | 3.54 | 8.78 | 4.84 | 8.20 | **9.23** | **18.35** | 21.46 | 29.82 |
| Wanda | ✗ | 4:8 | 2.65 | 7.40 | 3.10 | 8.13 | 6.60 | 10.84 | 12.87 | 20.92 |
| RIA | ✗ | 4:8 | 3.17 | 8.74 | 2.93 | 7.75 | 8.12 | 17.59 | 17.36 | 27.18 |
| MECON | ✗ | 4:8 | **3.71** | **9.29** | **4.95** | **8.53** | 8.38 | 17.59 | **21.80** | **30.39** |
| Magnitude | ✗ | 2:4 | 0.74 | 2.29 | 0.98 | 3.60 | 0.24 | 3.12 | 3.80 | 9.26 |
| SparseGPT | ✓ | 2:4 | 3.28 | **6.27** | 3.10 | 6.53 | 1.71 | **8.21** | 7.52 | 19.45 |
| Wanda | ✗ | 2:4 | 2.75 | 6.12 | 2.75 | 6.48 | 2.27 | 3.51 | 4.93 | 10.79 |
| RIA | ✗ | 2:4 | 2.56 | 4.73 | 2.79 | 5.65 | 1.98 | 6.74 | 6.49 | 17.22 |
| MECON | ✗ | 2:4 | **3.34** | **6.27** | **3.41** | **6.72** | **2.52** | 6.74 | **7.91** | **20.33** |

**Comparison to Pruner-Zero**  Pruner-Zero (Dong et al., 2024) is also an adaptation-based pruning method, which searches symbolic pruning metrics using genetic programming. Notably, MECON differs from Pruner-Zero in two key aspects: 1) Search Space: Pruner-Zero's search space involves weights, activations, and gradients, while MECON deliberately omits gradient computations. Despite this omission, as shown in Tables 2 and 4, Pruner-Zero even underperforms when compared to the baseline methods like Wanda and RIA, which rely on static metrics derived from weights and activations. Moreover, the calculation of gradients also introduces additional computational overhead. 2) Search Evaluation: Pruner-Zero uses perplexity on WikiText as search evaluation, whereas MECON relies on model-wise reconstruction error, thus substantially decreasing the evaluation duration. For instance, pruning LLaMA2-7B takes less than 10 seconds per trial with MECON, compared to over 70 seconds with Pruner-Zero.

**N:M Semi-Structured Pruning**  While MECON is designed for unstructured sparsity, it can be easily extended to semi-structured N:M sparsity (Mishra et al., 2021), which can leverage NVIDIA's sparse tensor cores to accelerate matrix multiplication in practice. In Table 5, we report the performance of 4:8 and 2:4 sparsity constraints on the WikiText and GMS8K datasets. We find that MECON consistently achieves superior performance than baselines, except LLaMA-3 models. We think this is because the LLaMA-3 model, trained on a larger amount of data, exhibits higher knowledge density (Meta, 2024). Thus, pruning a continuous block of parameters in semi-structured pruning leads to a significant performance drop, necessitating weight updates in SparseGPT for recovery.

Table 6: Pruning speed for pruning LLaMA-2/3 and Mistral models to 50% sparsity.

| Method | L2-7B | L2-13B | L3-8B | M-7B |
|---|---|---|---|---|
| SparseGPT | 370.03 | 464.77 | 457.71 | 450.76 |
| MECON | **56.16** | **107.11** | **60.11** | **59.80** |

Table 7: Inference speedup of different sparsity patterns for LLaMA-2/3 and Mistral models.

| Sparsity | L2-7B | L2-13B | L3-8B | M-7B |
|---|---|---|---|---|
| 4:8 | 1.11× | 1.04× | 1.15× | **1.17×** |
| 2:4 | **1.35×** | **1.14×** | 1.15× | 1.16× |

## 4.3 SPEEDUP

The theoretical computational complexity of SparseGPT is $O(d_{hidden}^3)$, while our meta pruning metric has a lower complexity of $O(d_{hidden}^2)$. We compare their empirical pruning speed on NVIDIA RTX A6000 GPUs by measuring the total time required to prune the model to 50% sparsity using each metric. Calibration data from the C4 training dataset is used to estimate activation magnitudes for the language modeling task. As shown in Table 6, our meta pruning metric results in negligible time overhead compared to SparseGPT. We further evaluate the inference speedup for semi-structured 4:8 and 2:4 sparsity on NVIDIA RTX A6000 GPUs. Our simulations utilize the high-performance

GEMM kernel from the NVIDIA CUTLASS library. According to the results presented in Table 7, when compared with dense models, we observe an average speedup of $1.20\times$ in end-to-end latency.

## 5 IN-DEPTH ANALYSIS

### 5.1 GENERALIZABILITY OF THE SEARCHED PRUNING METRICS

A potential limitation of MECON might be the relatively high cost of the search process, which is conventionally necessitated for every model across all datasets. To alleviate this problem, we explore the generalizability of our searched pruning metrics. We also outline the search costs for finding optimal metrics and layerwise sparsity ratios for LLaMA-1/2/3 and Mistral models in Appendix A.3.

**Cross-task Generalization.** We evaluate the generalizability of the pruning metrics identified from the complex arithmetic reasoning task (e.g. GSM8K) to the easier tasks, such as language modeling (e.g. WikiText) and zero-shot reasoning (e.g. LM Harness). This evaluation is partially inspired by the work (Fu et al., 2022) of multi-step reasoning which finds that complex demonstrations provide more valuable information. For instance, on the evaluation of LLaMA-1 7B on WikiText, we directly leverage the metrics searched with LLaMA-1 7B on GSM8K. As shown in Table 8, GSM8K metric, derived from the arithmetic reasoning task, consistently achieves the better performance than SparseGPT and comparable scores against the task-specific pruning metrics across LLaMA-1/2/3 and Mistral models on the WikiText and LM Harness benchmarks. We also conduct the counter experiment – performing the metrics derived from WikiText on the harder tasks, like LM Harness and GSM8K. We regrettably observe consistent performance declines on the target tasks. These findings suggest that the generalization works from the complex tasks to those easier.

**Cross-model Generalization.** For cross-model evaluation, we select models notable for their superior performance on arithmetic reasoning, specifically the LLaMA-2 7B and LLaMA-3 8B models. Metrics derived from these models, termed the LLaMA-2 and LLaMA-3 metrics, are applied across different model families to assess their effectiveness. The LLaMA-3 metric is tested on Mistral models, while the LLaMA-2 metric is evaluated with the LLaMA-1 models. We also tried the cross-model evaluations of transferring the metrics of LLaMA-3 to LLaMA-1/2 models. But we found the results are insignificant and we think the reason is that their weight distributions largely differ. As shown in Table 8, we find that the metrics from the superior model can consistently surpass the established baselines across the board. More excitingly, it even exceeds the original MECON. Further details of the optimal metrics found for each LLM, using both C4 and GSM8K calibration data, are provided in Appendix A.5.

Table 8: WikiText perplexity and zero-shot reasoning accuracy (%) with different pruning metrics.

| Method | WiKiText | | | | LM Harness | | | |
|---|---|---|---|---|---|---|---|---|
| | L1-7B | L2-7B | L3-8B | M-7B | L1-7B | L2-7B | L3-8B | M-7B |
| SparseGPT | 6.92 | 6.59 | 10.89 | 6.42 | 54.86 | 55.90 | 53.87 | 57.49 |
| MECON | 6.78 | **6.35** | **9.23** | 6.22 | 55.10 | **57.47** | 55.50 | **59.33** |
| GSM8K Metric | 6.78 | 6.39 | 12.78 | 6.23 | 55.15 | 56.05 | **55.59** | 57.66 |
| LLaMA-2 Metric | **6.76** | - | - | - | **55.24** | - | - | - |
| LLaMA-3 Metric | - | - | - | **6.16** | - | - | - | 58.30 |

Therefore, although we still claim the necessity of adaptive pruning for different models, we also provide a cost-effective alternative to mitigate the search process, which is adopting the pruning metric identified on the challenging task with the strongest model in your candidate pool. This metric has demonstrated a capacity for generalization, proving transferrable and reusable across less complex tasks or the less-performing models.

### 5.2 EFFECTIVENESS OF THE SEARCHED LAYERWISE SPARSITY RATIOS

Motivated by the distinct importance of parameters across different layers (Wang & Tu, 2020; Zhang et al., 2021), another component of MECON involves setting specific pruning ratios for each layer while maintaining an overall 50% reduction in parameters. Table 9 demonstrates that these layer-specific sparsity ratios, optimized through our pruning metric, not only enhance our model's performance but also significantly improve other baseline metrics, such as Wanda and RIA. Notably, these searched sparsity ratios lead to an average relative improvement of 4.68% in perplexity reduction

on the WikiText set. Furthermore, in LLaMA-3 models, these ratios enhance the performance of Wanda and RIA by an impressive score of 13.68%. Details of the layerwise sparsity ratios for each LLM are provided in Appendix A.5, and the results generally show that the upper layers tend to have more redundant parameters than the lower layers, aligning with findings from previous studies.

Table 9: Our searched layerwise sparsity ratios are effective for both Wanda and RIA metrics. The number (%) in (·) denotes the relative improvement (RI). For instance, Wanda RI = (Wanda w/ Ratio - Wanda) / Wanda.

| Method | Uniform | LLaMA-1 | | LLaMA-2 | | LLaMA-3 | | Mistral | |
|---|---|---|---|---|---|---|---|---|---|
| | | 7B | 13B | 7B | 13B | 8B | 8B-Inst | 7B | 7B-Inst |
| Wanda | ✓ | 6.90 | 5.82 | 6.47 | 5.64 | 10.57 | 16.37 | 7.24 | 7.22 |
| Wanda w/ Ratio | ✗ | 6.72 | 5.64 | 6.28 | 5.52 | 9.45 | 13.67 | 6.97 | 6.98 |
| | | (+2.61) | (+3.09) | (+2.94) | (+2.13) | (+10.60) | (+16.49) | (+3.73) | (+3.32) |
| RIA | ✓ | 6.81 | 5.83 | 6.43 | 5.63 | 12.56 | 15.57 | 7.27 | 7.21 |
| RIA w/ Ratio | ✗ | 6.65 | 5.67 | 6.26 | 5.54 | 10.98 | 13.23 | 6.89 | 6.96 |
| | | (+2.35) | (+2.74) | (+2.64) | (+1.60) | (+12.58) | (+15.03) | (+5.23) | (+3.47) |
| MECON wo/ Ratio | ✓ | 6.75 | 5.75 | 6.32 | 5.52 | 9.23 | 11.37 | 6.22 | 6.55 |
| MECON | ✗ | 6.61 | 5.60 | 6.19 | 5.44 | 8.95 | 10.73 | 6.08 | 6.39 |
| | | (+2.07) | (+2.61) | (+2.06) | (+1.45) | (+3.03) | (+5.63) | (+2.25) | (+2.62) |

## 5.3 SPARSITY

In Figure 3(a), we investigate the impact of different sparsity ratios, which range from 0.1 to 0.6, on the performance of the LLaMA-2 13B model. The perplexity curve demonstrates that MECON (red curve) consistently surpasses SparseGPT, Wanda, and RIA across all tested sparsity levels. Especially at the sparsity ratio of 60%, MECON outperforms the baselines by a large margin, achieving a 10.52% relative improvement compared to RIA.

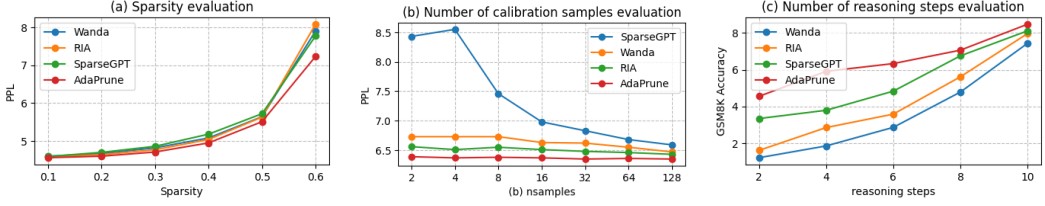

Figure 3: Sensitivity evaluation on sparsity, number of calibration samples (samples), and the reasoning steps in calibration samples for arithmetic reasoning.

## 5.4 ROBUSTNESS TO CALIBRATION SAMPLES.

We first vary the number of calibration samples for Wikitext evaluation on the LLaMA2-7B model. As illustrated in Figure 3(b), we see a clear difference in trend as the number of calibration samples ranging from 2 to 128. SparseGPT appears to rely on a larger number of calibration samples, while Wanda, RIA, and MECON are much more robust when there are few calibration samples. Notably, MECON consistently outperforms the other methods in all cases. We then vary the difficulty of calibration samples in arithmetic reasoning by selecting samples with reasoning steps ranging from 2 to 10. The results, summarized in Figure 3(c), clearly indicate that increasing the number of reasoning steps in calibration samples could improve the accuracy. MECON also consistently surpasses all baselines. Further ablation study, including an in-depth comparison and robustness analysis of the search algorithms and exploration of various search spaces, are provided in Appendix A.1 and A.2.

## 6 CONCLUSION

In this work, we introduce MECON, an adaptive pruning framework that automatically determines optimal pruning metrics and layerwise ratios for LLMs with diverse weight distributions. Inspired by the discovery of significant weight and activation features in LLMs, we create a meta pruning metric to balance these magnitudes. MECON identifies effective sparse networks in pretrained LLMs without retraining. Our evaluation shows that the metric from the best model for arithmetic reasoning also excels in simpler tasks with similar weight distributions.

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

# A APPENDIX / SUPPLEMENTAL MATERIAL

## A.1 ABLATION STUDY ON SEARCH ALGORITHMS AND ROBUSTNESS ANALYSIS

We conduct a robustness analysis using five search algorithms, including random search (Bergstra & Bengio, 2012), which randomly samples hyperparameter values from a predefined search space and does not take into account any information about the performance of previous trials, the Tree-structured Parzen Estimator (TPE) (Bergstra et al., 2011; 2013; Ozaki et al., 2022), a Bayesian optimization algorithm that uses a tree structure to model the relationship between hyperparameters and the objective function, and Quasi-Monte Carlo (QMC) (Bergstra & Bengio, 2012) sampler, which generates a sequence of points that cover the search space more evenly compared to random sampling, for more efficient exploration. Additionally, we utilize the state-of-the-art multi-objective optimization algorithms Non-dominated Sorting Genetic Algorithm II (NSGA-II) (Deb et al., 2002) and NSGA-III (Deb & Jain, 2013; Jain & Deb, 2013), which are based on genetic algorithms and optimize multiple conflicting objectives simultaneously. NSGA-II uses a non-dominated sorting approach to rank solutions based on their dominance relationship, while NSGA-III extends NSGA-II by incorporating reference points to guide the search toward the Pareto front.

Table 10: Statistical results of different search algorithms on LLaMA-2 7B model. We report the mean and standard deviation under 3 search process runs.

| Dataset | Random | TPE | QMC | NSGA-II | NSGA-III |
|---|---|---|---|---|---|
| WikiText | 6.89 ($\pm$ 0.0671) | **6.33** ($\pm$ 0.0714) | 6.39 ($\pm$ 0.0700) | 6.44 ($\pm$ 0.0632) | 6.35 ($\pm$ 0.0640) |
| GSM8K | 7.96 ($\pm$ 0.2406) | 8.33 ($\pm$ 0.2498) | 8.08 ($\pm$ 0.2220) | 8.47 ($\pm$ 0.2479) | **8.49** ($\pm$ 0.2646) |
| MMLU | 31.11 ($\pm$ 0.3962) | 31.06 ($\pm$ 0.4017) | 31.80 ($\pm$ 0.4400) | 32.43 ($\pm$ 0.4701) | **33.06** ($\pm$ 0.4687) |
| LM-harness | 55.32 ($\pm$ 0.5300) | 55.74 ($\pm$ 0.5367) | 56.19 ($\pm$ 0.5234) | 56.59 ($\pm$ 0.5689) | **57.47** ($\pm$ 0.5718) |

Table 10 presents the statistical analysis, specifically the mean and standard deviation, of the performance of pruned LLaMA-2 7B models across four distinct benchmarks. To validate the robustness and reliability of our results, each model was evaluated using three different search processes, each initialized with different random seeds. We report the performance outcomes of the NSGA-III search method in the main paper, as it generally outperforms other algorithms.

## A.2 ABLATION STUDY ON SEARCH SPACE

In Table 11, we construct the sub-search spaces by randomly selecting 2-6 coefficients/operations from our candidate pool. We test three different subspaces using random seeds 0, 42, and 100. The evaluations are conducted on WikiText, and the perplexity scores are reported below. We can see that the search performed on the full set consistently yields the best results. An interesting observation is that using a very small subspace may lead to extremely poor outcomes. This occurs because the candidate coefficients/operations in the subspace are all unsuitable for the target model.

Table 11: WikiText perplexity for random subspaces of the search space on LLaMA2-7B model in searching for optimal pruning metric.

| Seed | 2 Ops & 2 Coes | 3 Ops & 3 Coes | 4 Ops & 4 Coes | 5 Ops & 5 Coes | 6 Ops & 6 Coes | Full Search Space |
|---|---|---|---|---|---|---|
| 0 | 870.16 | 753.32 | 6.43 | 6.51 | 6.57 | **6.35** |
| 42 | 6.47 | 6.39 | 6.39 | 6.39 | 6.35 | **6.35** |
| 100 | 6.43 | 6.43 | 6.43 | 6.58 | 6.58 | **6.35** |

## A.3 SEARCH COST

In Table 12, we provide the detailed search time consumed on a single Nvidia RTX A6000 GPU. We report the total time of 350 search trials, as we empirically found that the optimal value is generally obtained within these rounds, as illustrated in Figure 4(c). As shown in the table, the time of an optimal search is within 2.5 GPU hours. With multiple GPUs, the search process can generally be finished within 1 hour. Therefore, we believe that the search cost of our method is moderate and acceptable.

Table 12: Cost of searching for optimal pruning metric and layerwise sparsity ratios on LLaMA-1/2/3 and Mistral models.

| Search | L1-7B | L1-13B | L2-7B | L2-13B | L3-8B | L3-8B-it | M-7B | M-7B-it |
|---|---|---|---|---|---|---|---|---|
| Metric | 1h10m28s | 2h13m6s | 1h6m14s | 2h11m55s | 1h30m47s | 1h31m51s | 1h14m22s | 1h15m54s |
| Ratio | 1h13m44s | 2h19m28s | 1h9m34s | 2h22m45s | 1h31m59s | 1h32m51s | 1h17m8s | 1h18m12s |

### A.4 HYPERPARAMETER ANALYSIS

We evaluate the impact of various hyperparameters applied in the layerwise sparsity ratios search procession the performance of WikiText perplexity. We use the LLaMA-2 7B model and prune to unstructured 50% sparsity.

**Sparsity step.** In layerwise sparsity ratio search, we identify the optimal sparsity ratio for each layer by selecting from a predefined sparsity ratio set: [target sparsity - sparsity step, target sparsity, target sparsity + sparsity step]. Here, target sparsity is the pre-defined sparsity ratio for pruning the overall model. The sparsity step allows for adjustments to achieve slightly higher or lower sparsity ratios.

In Figure 4(a), we vary the sparsity step ranging between 3% and 10%. We empirically find that a 5% sparsity step usually performs better than other sparsity steps, such as lower 3% or higher 8% and 10%. This is possibly because smaller steps might not significantly reduce redundancy, while larger steps might overly simplify the layers, leading to a loss of important features and a decrease on overall model performance.

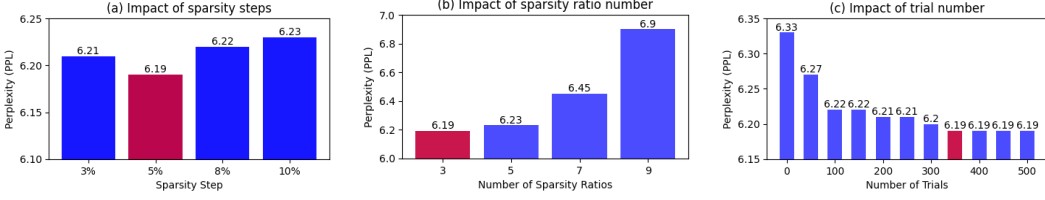

Figure 4: Sensitivity evaluation on sparsity step, number of sparsity ratios, and the number of trials in layerwise sparsity ratio search.

**Number of sparsity ratios.** In Figure 4(b), we fix the sparsity step as 5% and vary the number of sparsity ratios in the predefined sparsity ratio set, which ranges from 3 to 9. Specifically, one sparsity ratio in the sparsity ratio set corresponds to uniform pruning across layers. For example, a predefined sparsity ratio set with 5 sparsity ratios is defined as [target sparsity - 2*sparsity step, target sparsity - sparsity step, target sparsity, target sparsity + sparsity step, target sparsity + 2*sparsity step].

We empirically find that, for LLaMA2-7B model that contains 32 layers, a discrete sparsity set of 3 sparsity ratios is able to search for better results than larger sets of sparsity ratios. This possibly because a larger number of sparsity ratios significantly expands the search space, making it challenging to find the optimal solution within a limited number of search trials.

**Number of search trials.** In Figure 4(c), we investigate the influence of varying the number of trials on the performance of the NSGA-III search algorithm. The trials evaluated range from an initial count of 0 up to a maximum of 500. The results reveals that the perplexity stabilizes and reaches an optimal value at the point where the number of search trials is set to 350. Based on this empirical evidence, we select this specific number of trials for the NSGA-III algorithm in our experiments discussed in the main paper.

## A.5 Optimal Pruning Metric and Layerwise Sparsity Ratios

Our meta pruning metric adjusts the relationship between weight and activation magnitudes by applying specific coefficients and operations to both weight and activation magnitudes. The operation sets include (1) no op, which leaves the matrix unchanged, (2) sqrt, which computes the square root of each matrix element, and (3) square, which raises each element to the power of two. The coefficient sets include (1) no coe, which leaves the scaling of the matrix elements unchanged, (2) F norm, using the reciprocal of the Frobenius norm of the matrix, (3) to sum, and (4) to mean, setting the coefficients as the reciprocal of the total sum and the average of the matrix elements, respectively. (5) row sum and (6) column sum, using the reciprocal of the sums of specific rows or columns, respectively. Finally, (7) relative sum calculates coefficients as the sum of the row sums and column sums for each matrix element. The detailed calculation equations are illustrated in Table 13, using matrix $A = A_{ij}$ with $m$ rows and $n$ columns as input for demonstration.

The detailed calculations for the coefficient sets utilized in our pruning metric are comprehensively illustrated in Table 13. For these calculations, we use a matrix $A = A_{ij}$ that consists of $m$ rows and $n$ columns as input demonstration.

Table 13: Detailed calculations for the coefficient sets in meta pruning metric.

| coefficient | Equation | coefficient | Equation |
|---|---|---|---|
| no coe | $\alpha = \beta = 1$ | F norm | $\alpha = \beta = 1/\sqrt{\sum_{i=1}^{m}\sum_{j=1}^{n} A_{ij}^2}$ |
| to sum | $\alpha = \beta = 1/\sum_{i=1}^{m}\sum_{j=1}^{n} A_{ij}$ | to mean | $\alpha = \beta = mn/\sum_{i=1}^{m}\sum_{j=1}^{n} A_{ij}$ |
| row sum | $\alpha = \beta = 1/\sum_{j=1}^{n} A_{ij}$ | column sum | $\alpha = \beta = 1/\sum_{i=1}^{m} A_{ij}$ |
| relative sum | $\alpha = \beta =$ row sum $(A_{ij})+$ column sum $(A_{ij})$ | | |

**Optimal pruning metrics.** In Table 14, we present the optimal coefficients and operations for pruning metrics using samples from the C4 dataset as calibration data. Table 15 displays the optimal coefficients and operations for pruning metrics using samples from the GSM8K dataset as calibration data. Compared to the results based on the C4 dataset, the metrics derived from the GSM8K dataset show a greater divergence from RIA metric (Zhang et al.). Notably, most of these metrics do not incorporate the relative sum as a weight coefficient.

Table 14: Optimal coefficients and operations for pruning metrics on C4 calibration data.

| Metric | LLaMA-1 | | LLaMA-2 | | LLaMA-3 | | Mistral | |
|---|---|---|---|---|---|---|---|---|
| | 7B | 13B | 7B | 13B | 8B | 8B-Inst | 7B | 7B-Inst |
| $\alpha$ | relative sum | relative sum | relative sum | relative sum | relative sum | no coe | relative sum | F norm |
| $\beta$ | to mean | to mean | to mean | no coe | F norm | F norm | to mean | to mean |
| $\tau_1$ | no op | square | no op | square | no op | no op | square | sqrt |
| $\tau_2$ | sqrt | no op | sqrt | sqrt | no op | no op | no op | sqrt |

Table 15: Optimal coefficients and operations for pruning metrics on GSM8K calibration data.

| Metric | LLaMA-1 | | LLaMA-2 | | LLaMA-3 | | Mistral | |
|---|---|---|---|---|---|---|---|---|
| | 7B | 13B | 7B | 13B | 8B | 8B-Inst | 7B | 7B-Inst |
| $\alpha$ | row sum | to mean | F norm | column sum | to mean | relative sum | row sum | relative sum |
| $\beta$ | relative sum | F norm | to sum | relative sum | to sum | no coe | no coe | to mean |
| $\tau_1$ | no op | no op | no op | square | square | no op | square | square |
| $\tau_2$ | sqrt | sqrt | sqrt | sqrt | sqrt | no op | no op | no op |

**Optimal layerwise pruning ratio.** In Figure 5, we report the optimal layerwise sparsity ratios for LLaMA-1/2/3 and Mistral models. The results generally indicate that the upper layers contain more redundant parameters compared to the lower layers, as higher sparsity ratios are more common in the top layers, while lower sparsity ratios are more frequent in the lower layers.

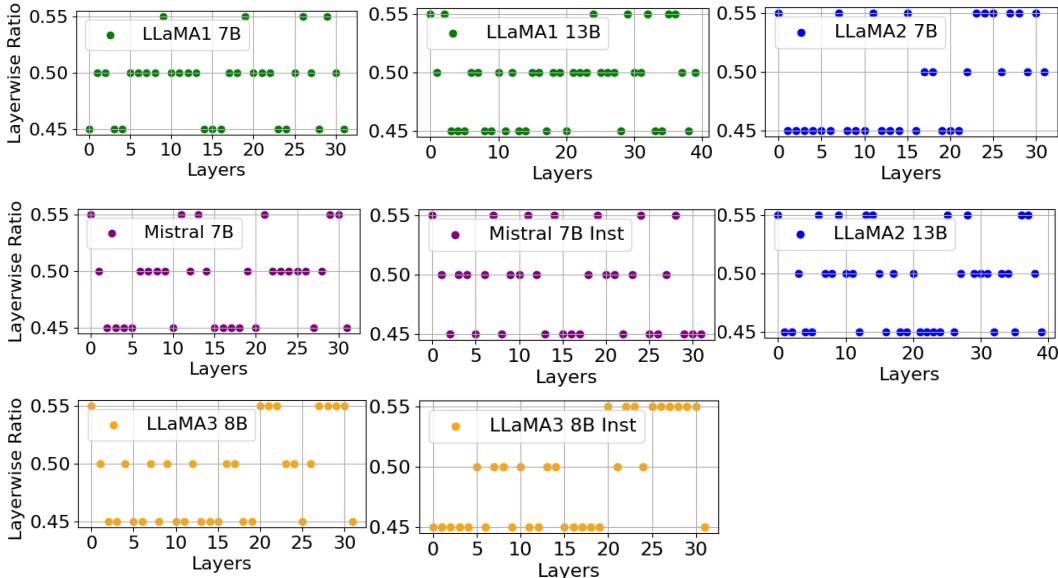

Figure 5: Searched layerwise sparsity ratios for LLaMA-1/2/3 and Mistral models.

## A.6 Relationship between Transformed Wights and Activations

In our analysis of the optimal searched pruning metrics, We find that the differences between transformed weights and transformed activations may affect the effectiveness of different pruning metrics. Specifically, we analyze each pruning metric, such as Wanda, RIA, and Mecon, by decomposing them into two distinct components: the transformed weights and the transformed activations, each defined by specific coefficients or operations. As the SparseGPT metric combines weights and the Hessian matrix, and the Wanda metric serves as a simpler approximation of the SparseGPT metric. Due to this relationship, we omit the weight and activation analysis for SparseGPT.

We measure the difference between transformed weights and activations as the layer-wise absolute difference, which is calculated by summing the average absolute differences across all linear submodules in each layer. We report the average layer-wise differences between the operated weights and the operated activations across the Wanda, RIA, and Mecon pruning metrics in Table 16, with detailed layer-wise difference curves available in Figures 6, 7, 8, 9.

Table 16: Average absolute difference between operated weights and operated activations for Wanda, RIA and Mecon on C4 Calibration Data.

| Method | LLaMA-2 7B | LLaMA-2 13B | LLaMA-3 8B | Mistral 7B |
|--------|-----------|-------------|------------|------------|
| Wanda  | 82.66     | 78.30       | 79.31      | 392.13     |
| RIA    | 22.34     | 21.91       | 21.15      | 44.77      |
| Mecon  | 1.09      | 0.0001      | 0.1263     | 0.0304     |

Table 16 shows that the RIA pruning metric reduces the absolute difference compared to Wanda, while the Mecon searched metric further minimizes this difference, bringing it close to zero. The weighted transformation operation in the Mecon pruning metric effectively scales both weights and activations into a similar numerical range, facilitating a balanced evaluation of each weight relative to its corresponding activation.

Coupled with the performance results of each pruning metric presented in Table 2, the difference analysis in Table 1 suggests that pruning metrics with smaller absolute differences between transformed weights and activations are more likely to achieve effective pruning. Thus, the performance of Wanda and other methods may be influenced by how well they account for these differences regarding different models with different weight magnitudes and distributions.

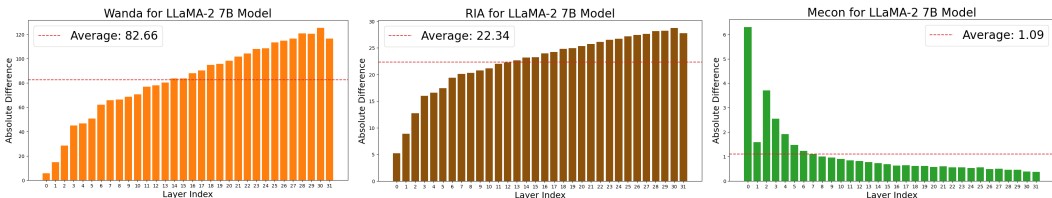

Figure 6: Layerwise absolute distance between transformed weights and transformed activations for Wanda, RIA, and Mecon metrics on LLaMA-2 7B models.

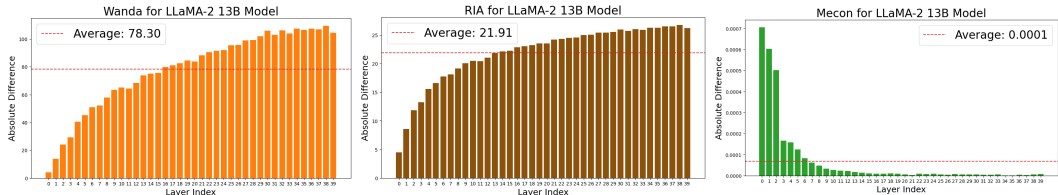

Figure 7: Layerwise absolute distance between transformed weights and transformed activations for Wanda, RIA, and Mecon metrics on LLaMA-2 13B models.

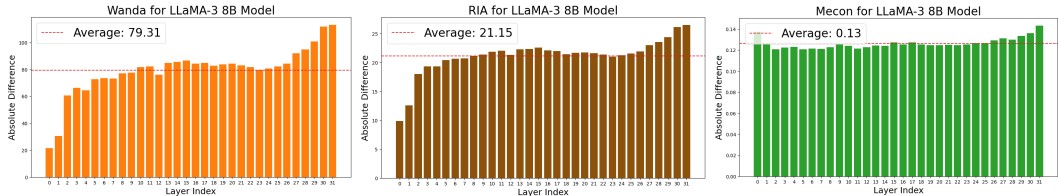

Figure 8: Layerwise absolute distance between transformed weights and transformed activations for Wanda, RIA, and Mecon metrics on LLaMA-3 8B models.

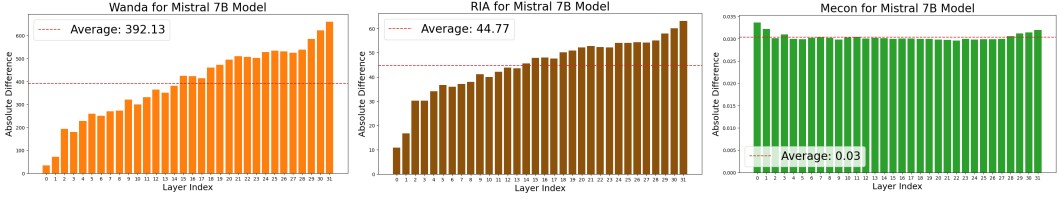

Figure 9: Layerwise absolute distance between transformed weights and transformed activations for Wanda, RIA, and Mecon metrics on Mistral 7B models.

## A.7 TASK-WISE RESULTS ON LM HARNESS

For LM-harness results, the 7 evaluated zero-shot tasks are: BoolQ (Clark et al., 2019), RTE (Wang et al., 2018), HellaSwag (Zellers et al., 2019), WinoGrande (Sakaguchi et al., 2021), ARC Easy and Challenge (Clark et al., 2018), and OpenbookQA (Mihaylov et al., 2018). For reproducibility, we used v0.4.0 release. All tasks were evaluated on task version 0 except for BoolQ on task version 1. We show the task-wise performance of mean zero-shot accuracies of pruned LLaMA-1/2/3 and Mistral models in Tables 17, 18, 19, 20, 21, 22, 23, 24.

Table 17: Accuracies (%) of LLaMA-1 7B model for 7 zero-shot tasks with unstructured 50% sparsity.

| Method | BoolQ | RTE | HellaSwag | WinoGrande | ARC-e | ARC-c | OBQA | Average |
|---|---|---|---|---|---|---|---|---|
| Dense | 75.06 | 66.23 | 56.93 | 69.54 | 74.82 | 41.02 | 34.30 | 59.70 |
| Magnitude | 55.10 | 54.51 | 45.49 | 59.10 | 58.65 | 32.97 | 22.40 | 46.89 |
| SparseGPT | 72.03 | 54.15 | 51.43 | 67.87 | 71.39 | 37.54 | 29.60 | 54.86 |
| Wanda | 71.04 | 54.51 | 51.93 | 65.90 | 69.40 | **36.95** | 28.80 | 54.08 |
| RIA | 72.84 | 57.76 | 51.93 | 66.85 | **70.50** | 36.43 | 29.40 | 55.10 |
| Pruner-Zero | 70.28 | 56.68 | 47.27 | 64.96 | 66.92 | 33.25 | 26.80 | 52.31 |
| Our Metric | **72.87** | 57.40 | 51.91 | 67.25 | 70.33 | 36.35 | 29.60 | 55.10 |
| GSM8K Metric | 71.04 | **58.48** | 52.39 | 67.17 | 69.91 | 37.46 | **30.20** | **55.24** |
| LLaMA2 Metric | 70.73 | 57.63 | **53.24** | 67.01 | 70.24 | 37.97 | **30.20** | 55.15 |

Table 18: Accuracies (%) of LLaMA-1 13B model for 7 zero-shot tasks with unstructured 50% sparsity.

| Method | BoolQ | RTE | HellaSwag | WinoGrande | ARC-e | ARC-c | OBQA | Average |
|---|---|---|---|---|---|---|---|---|
| Dense | 78.03 | 70.51 | 59.63 | 72.89 | 77.28 | 46.55 | 33.20 | 62.58 |
| Magnitude | 55.19 | 52.23 | 43.65 | 63.36 | 57.82 | 32.53 | 26.60 | 47.34 |
| SparseGPT | **76.89** | 60.95 | 54.99 | 71.46 | 72.15 | 42.17 | 31.20 | 58.54 |
| Wanda | 75.73 | 62.48 | 55.70 | 71.68 | 72.91 | 43.45 | 32.20 | 59.18 |
| RIA | 76.44 | 62.34 | 56.13 | 72.73 | 72.42 | **43.87** | 32.20 | 59.45 |
| Pruner-Zero | 73.91 | 62.36 | 52.65 | 69.41 | 70.83 | 41.62 | 28.80 | 57.08 |
| Our Metric | 76.67 | 62.45 | 56.11 | **73.63** | 73.25 | 43.62 | **32.40** | **59.73** |
| GSM8K Metric | 76.62 | **62.89** | 55.48 | 72.79 | 72.58 | 43.78 | 32.00 | 59.45 |
| LLaMA2 Metric | 76.51 | 62.32 | **56.43** | 71.82 | **73.39** | 43.84 | **32.40** | 59.53 |

Table 19: Accuracies (%) of LLaMA-2 7B model for 7 zero-shot tasks with unstructured 50% sparsity.

| Method | BoolQ | RTE | HellaSwag | WinoGrande | ARC-e | ARC-c | OBQA | Average |
|---|---|---|---|---|---|---|---|---|
| Dense | 77.74 | 62.82 | 57.14 | 69.14 | 76.35 | 43.43 | 31.40 | 59.72 |
| Magnitude | 62.57 | 52.35 | 52.99 | 65.35 | 67.97 | 37.20 | 28.40 | 52.40 |
| SparseGPT | **75.78** | 57.75 | 52.90 | **69.14** | 71.34 | 37.97 | 26.60 | 55.90 |
| Wanda | 75.35 | 53.43 | 52.63 | 67.25 | **72.35** | 39.42 | 30.80 | 55.89 |
| RIA | 75.66 | 53.79 | 52.25 | 67.25 | 72.05 | 37.71 | **31.00** | 55.67 |
| Pruner-Zero | 73.48 | 53.29 | 49.18 | 65.83 | 69.92 | 38.36 | 26.60 | 53.81 |
| Our Metric | 74.62 | **62.82** | **57.14** | 68.03 | 71.00 | 38.91 | 29.80 | **57.47** |
| GSM8K Metric | 75.11 | 53.79 | 53.55 | 67.25 | 72.31 | **39.93** | 30.40 | 56.05 |
| LLaMA2 Metric | 75.11 | 53.79 | 53.55 | 67.25 | 72.31 | **39.93** | 30.40 | 56.05 |

Table 20: Accuracies (%) of LLaMA-2 13B model for 7 zero-shot tasks with unstructured 50% sparsity.

| Method | BoolQ | RTE | HellaSwag | WinoGrande | ARC-e | ARC-c | OBQA | Average |
|---|---|---|---|---|---|---|---|---|
| Dense | 80.52 | 65.34 | 60.33 | 71.95 | 79.38 | 48.47 | 35.20 | 63.03 |
| Magnitude | 57.62 | 55.87 | 54.53 | 65.85 | 70.47 | 38.13 | 27.80 | 52.90 |
| SparseGPT | 81.42 | 65.26 | 55.83 | 72.64 | 74.91 | 42.23 | **32.60** | 60.70 |
| Wanda | 81.86 | 64.08 | 56.92 | 71.37 | 76.12 | 43.81 | 32.00 | 60.88 |
| RIA | **81.93** | 64.02 | 57.73 | 71.89 | 76.24 | 43.46 | 32.00 | 61.03 |
| Pruner-Zero | 77.86 | 61.22 | 56.89 | 67.90 | 74.16 | 39.81 | 29.40 | 58.18 |
| Our Metric | 80.97 | **66.17** | 59.68 | **72.35** | 76.29 | 43.68 | 30.80 | 61.42 |
| GSM8K Metric | 81.56 | 64.06 | 58.41 | 72.23 | 76.98 | 43.73 | 32.00 | 61.28 |
| LLaMA2 Metric | 80.25 | 66.14 | **59.73** | 71.57 | **77.36** | **43.85** | 32.00 | **61.56** |

Table 21: Accuracies (%) of LLaMA-3 8B model for 7 zero-shot tasks with unstructured 50% sparsity.

| Method | BoolQ | RTE | HellaSwag | WinoGrande | ARC-e | ARC-c | OBQA | Average |
|---|---|---|---|---|---|---|---|---|
| Dense | 81.44 | 69.68 | 60.17 | 72.85 | 80.09 | 50.43 | 34.80 | 64.21 |
| Magnitude | 49.14 | 53.43 | 38.55 | 55.09 | 60.69 | 32.42 | 24.80 | 44.87 |
| SparseGPT | 74.80 | **54.15** | **49.90** | 68.35 | 67.05 | 36.43 | 26.40 | 53.87 |
| Wanda | 73.43 | 52.71 | 41.80 | 63.22 | 64.86 | 29.78 | 21.80 | 49.66 |
| RIA | 75.20 | 53.12 | 43.00 | 64.56 | 65.87 | 30.55 | 23.00 | 50.76 |
| Pruner-Zero | 72.32 | 54.51 | 45.78 | 65.19 | 70.58 | 35.41 | 23.60 | 52.48 |
| Our Metric | **79.54** | 53.07 | 43.24 | **70.24** | 72.05 | 41.13 | **29.20** | 55.50 |
| GSM8K Metric | 73.88 | **63.90** | 49.68 | 68.90 | 70.37 | 37.80 | 24.60 | **55.59** |
| LLaMA3 Metric | 73.88 | **63.90** | 49.68 | 68.90 | 70.37 | 37.80 | 24.60 | **55.59** |

Table 22: Accuracies (%) of Instruction-tuned LLaMA-3 8B model for 7 zero-shot tasks with unstructured 50% sparsity.

| Method | BoolQ | RTE | HellaSwag | WinoGrande | ARC-e | ARC-c | OBQA | Average |
|---|---|---|---|---|---|---|---|---|
| Dense | 83.06 | 67.51 | 57.68 | 71.98 | 81.61 | 52.99 | 34.20 | 64.15 |
| Magnitude | 68.84 | 60.65 | 36.31 | 53.75 | 49.83 | 26.19 | 21.80 | 45.31 |
| SparseGPT | 77.00 | 60.65 | **49.61** | 66.46 | 70.92 | 40.19 | 26.40 | 55.89 |
| Wanda | 76.57 | 54.51 | 41.18 | 63.61 | 67.63 | 33.70 | 22.20 | 51.34 |
| RIA | 78.17 | 54.51 | 42.29 | 64.25 | 68.35 | 34.13 | 22.80 | 50.64 |
| Pruner-Zero | 76.88 | 54.51 | 45.32 | 65.67 | 69.44 | 36.95 | 25.00 | 55.60 |
| Our Metric | **81.56** | 54.15 | 42.32 | **68.11** | 74.28 | 41.55 | 29.60 | **55.94** |
| GSM8K Metric | 78.17 | 54.51 | 42.29 | 64.25 | 68.35 | 34.13 | 22.80 | 50.64 |
| LLaMA3 Metric | 76.82 | **62.45** | 48.18 | 66.30 | 71.34 | 39.59 | 26.80 | 55.93 |

Table 23: Accuracies (%) of Mistral 7B model for 7 zero-shot tasks with unstructured 50% sparsity.

| Method | BoolQ | RTE | HellaSwag | WinoGrande | ARC-e | ARC-c | OBQA | Average |
|---|---|---|---|---|---|---|---|---|
| Dense | 81.44 | 69.68 | 60.17 | 72.85 | 80.09 | 50.43 | 34.80 | 64.21 |
| Magnitude | 75.87 | 55.60 | 56.74 | 68.35 | 74.20 | 42.15 | 27.80 | 57.24 |
| SparseGPT | 76.73 | **61.01** | 54.52 | 67.72 | 74.24 | 41.64 | 26.60 | 57.49 |
| Wanda | 76.12 | 55.60 | 48.95 | 65.59 | 72.69 | 37.46 | 23.00 | 54.20 |
| RIA | 76.48 | 56.68 | 49.05 | 66.30 | 72.47 | 37.12 | 22.60 | 54.39 |
| Pruner-Zero | 77.46 | 60.65 | 50.25 | 68.90 | 71.84 | 37.46 | 22.40 | 55.57 |
| Our Metric | **82.35** | 56.68 | 55.77 | **70.88** | 76.18 | 45.22 | 28.22 | **59.33** |
| GSM8K Metric | 81.53 | 55.60 | 54.43 | 69.38 | 74.16 | 42.15 | 26.40 | 57.66 |
| LLaMA3 Metric | 80.52 | 56.32 | **55.94** | 69.53 | 75.00 | 42.41 | **28.40** | 58.30 |

Table 24: Accuracies (%) of Instruction-tuned Mistral 7B model for 7 zero-shot tasks with unstructured 50% sparsity.

| Method | BoolQ | RTE | HellaSwag | WinoGrande | ARC-e | ARC-c | OBQA | Average |
|---|---|---|---|---|---|---|---|---|
| Dense | 83.06 | 67.51 | 57.68 | 71.98 | 81.61 | 52.99 | 34.20 | 64.15 |
| Magnitude | 79.09 | 65.06 | 59.31 | 67.43 | 77.96 | 49.77 | 31.80 | 62.34 |
| SparseGPT | 81.56 | **72.92** | 58.77 | 70.01 | 76.85 | 48.72 | 28.40 | 62.46 |
| Wanda | 83.73 | 66.79 | 55.68 | 67.48 | 77.06 | 48.12 | 28.40 | 61.04 |
| RIA | 83.88 | 66.79 | 55.61 | 67.32 | 77.78 | 47.95 | 27.60 | 60.48 |
| Pruner-Zero | 83.18 | 68.95 | 56.17 | 68.27 | 76.43 | 47.44 | 29.40 | 61.41 |
| Our Metric | **84.40** | 66.79 | 58.75 | **70.24** | 80.13 | 51.45 | 32.80 | **63.51** |
| GSM8K Metric | 84.59 | 67.87 | 58.97 | 68.90 | 78.11 | 51.11 | 31.80 | 63.05 |
| LLaMA3 Metric | 84.13 | 66.06 | **59.87** | 69.14 | 78.79 | 51.37 | 32.00 | 63.05 |

