# OpenReview forum: "Towards Efficient Adaptation of Pruning Strategy in Large Language Models"
_ICLR.cc/2025/Conference — Submitted to ICLR 2025_

### Official Review · Reviewer_Mjmt · 2024-10-27

**Soundness:** 2
**Presentation:** 2
**Contribution:** 1
**Rating:** 3
**Confidence:** 5

**Summary:**

1. The paper presents MECON, an evolutionary optimization framework designed for adaptive pruning of large language models (LLMs). Recognizing that fixed pruning strategies are ineffective due to the diverse weight distributions across LLMs, the authors develop a Meta pruning metric and an efficient search space to address this challenge.

2. The framework utilizes model-wise reconstruction error as a fast evaluation method and employs the NSGA-III algorithm to optimize both single-objective and multi-objective pruning problems. Extensive experiments on various LLaMA and Mistral models show that MECON's adaptive pruning approach outperforms existing methods, improves pruning efficiency, and demonstrates cross-task and cross-model generalizability.

**Strengths:**

1. The paper is clearly written and free of obvious typos.
2. The proposed evolutionary computation-based algorithm for searching hyperparameters in large model pruning is tested on well-known models such as LLaMA-1/2/3 and Mistral, as well as on datasets like WikiText, GSM8K, and MMLU.
3. The authors conduct an ablation study on the evolutionary computation algorithm used, attempting to justify the choice of Non-dominated Sorting Genetic Algorithm III (NSGA-III) as their search algorithm.

**Weaknesses:**

1.  The most critical issue with this paper is the lack of testing for the algorithm's efficiency. The authors only evaluate the accuracy under unstructured and semi-structured (N: M) pruning with a 50% mask compression rate across various datasets but do not provide results on memory efficiency or latency in GPU-CPU collaborative scenarios. Without these efficiency metrics, especially for unstructured pruning, it is unclear whether the method can be applied in real-world scenarios.

2. The paper only tests and compares baselines at a 50% compression rate and does not explore other extreme compression rates, such as below 50%.

3. The datasets used to validate the algorithm are limited, as testing is only conducted on GSM8K, MMLU, and WikiText. It would be beneficial to evaluate the model's generalizability on a broader range of datasets, such as those from the lm-evaluation-harness library.

4. The performance improvement of the algorithm appears marginal on some datasets and models. For instance, in Table-2, MECON achieves a score of 63.51 on the Mistral model, while the simpler Magnitude pruning achieves 63.34. This minor gain does not justify the significantly higher time and complexity involved in MECON’s search and pruning process compared to straightforward methods like Magnitude pruning.

5. Some relevant works on compression of LLMs using search algorithms/NAS methods are not mentioned or compared in the paper, such as:
[1] Pruning Large Language Models via Accuracy Predictor.
[2] Tune As You Scale: Hyperparameter Optimization For Compute Efficient Training.

**Questions:**

1. Can the authors conduct efficiency testing experiments?

2. Can the authors perform experiments at other compression rates?

3. Can the authors provide additional dataset evaluations to demonstrate the algorithm’s generalizability? Given that post-training algorithms typically do not consume significant computational resources or time, this should be feasible.

---

> ### Author Response · Authors · 2024-11-19
> **Response to Reviewer Mjmt**
>
> Thank you for your valuable feedback and for taking the time to review our work. We appreciate your recognition of our contributions.
>
> ---
>
> **Q1. Efficiency Testing Experiments.**
>
> We have conducted efficiency testing experiments, as detailed in Appendix A.3 of the paper. The results demonstrate the effectiveness of our proposed method in terms of both speed and resource utilization. We here present the table for your convenience.
>
> Table 1: Cost of Searching for Optimal Pruning Metric and Layerwise Sparsity Ratios on LLaMA-1/2/3 and Mistral Models
>
> | Search       | L1-7B      | L1-13B     | L2-7B      | L2-13B     | L3-8B      | L3-8B-it   | M-7B       | M-7B-it    |
> |--------------|------------|------------|------------|------------|------------|------------|------------|------------|
> | Metric       | 1h 10m 28s | 2h 13m 6s  | 1h 6m 14s  | 2h 11m 55s | 1h 30m 47s | 1h 31m 51s | 1h 14m 22s | 1h 15m 54s |
> | Ratio        | 1h 13m 44s | 2h 19m 28s | 1h 9m 34s  | 2h 22m 45s | 1h 31m 59s | 1h 32m 51s | 1h 17m 8s  | 1h 18m 12s |
>
> As shown in the table, the time of an optimal search is within 2.5 GPU hours. With multiple GPUs, the search process can generally be finished within 1 hour. Therefore, we believe that the search cost of our method is moderate and acceptable in real-world scenarios.
>
> ---
>
> **Q2.Experiments at Other Compression Rates.**
>
> Our experiments also include evaluations at various compression rates, which are presented in Figure 3(a). These results illustrate how our method performs under different levels of sparsity ranging from 10% and 60%.
>
> Table 2: Comparison of pruning metrics across different compression rates on WikiText perplexity.
>
> | Method       | 10%  | 20%  | 30%  | 40%  | 50%  | 60%  |
> |--------------|------|------|------|------|------|------|
> | Wanda        | 4.60 | 4.67 | 4.82 | 5.08 | 5.64 | 7.90 |
> | RIA          | 4.58 | 4.64 | 4.77 | 5.04 | 5.63 | 8.08 |
> | SparseGPT    | 4.59 | 4.70 | 4.86 | 5.18 | 5.72 | 7.77 |
> | Mecon        | 4.56 | 4.60 | 4.71 | 4.95 | 5.51 | 7.23 |
>
> The perplexity results consistently demonstrate that our Mecon method outperforms the baseline approaches across all the tested sparsity levels. Notably, at the challenging 60% sparsity ratio, Mecon exhibits a significant performance advantage. We hope this clarifies any oversight of our experiment results.
>
> ---
>
> **Q3. Additional Dataset Evaluations.**
>
> We have also evaluated our method on 7 datasets from LM-Harness to showcase its generalizability, including BoolQ, RTE, HellaSwag, WinoGrande, ARC-easy, ARC-challenge and OpenbookQA. These evaluations can be found in Section 4.2, Table 2 and Table 3, where we discuss the performance across different datasets from LM-Harness. Besides, we report the detailed task-wise accuracy on LM Harness in Appendix A.7.
>
> ---
>
> **Q4. Performance Improvements from Magnitude Pruning.**
>
> As you noted, magnitude pruning performs well on the LM-Harness tasks for the Mistral 7B model, even exceeding the performance of activation-considered pruning metrics like SparseGPT, Wanda, and RIA. The improvement seen with Mecon may appear marginal. However, Mecon's benefits extend beyond individual metrics; its adaptive approach maintains effectiveness across a broader range of models and tasks.
>
> A deeper understanding lies in the performance differences between magnitude pruning and activation-considered pruning metrics across various models and tasks.
>
> - Magnitude pruning evaluates weight importance solely based on absolute values, making it more robust to shifts in calibration data distribution, as it does not depend on specific data patterns for pruning decisions.
>
> - Activation-considered pruning methods, like SparseGPT, Wanda, and RIA, assess weight importance based on activations relative to calibration data. If the calibration data differs significantly from the distribution during model pretraining, these methods may misestimate the importance of certain weights.
>
> Consequently, when using the same calibration data across different models (e.g., LLaMA and Mistral), magnitude pruning and activation-considered methods can behave differently.
>
> ---
>
> We hope our response addresses your concerns and provides clarity on the effectiveness of our proposed method. Thank you once again for your valuable feedback!

---

> > ### Author Response · Authors · 2024-11-25
> > **Free to any further discussions**
> >
> > Dear Reviewer Mjmt,
> >
> > Thank you for taking the time to review our submission and provide valuable feedback. As mentioned in our previous response, the evaluations on efficiency, various compression rates, and the use of the lm-evaluation-harness library were already included in our initial manuscript. Additionally, we have provided a more in-depth discussion on the observed performance differences. We sincerely appreciate your comments regarding the missing relevant works and will ensure they are incorporated into our final revision.
> >
> > As the deadline of discussion period is apporaching, we would really appreciate it if you have read our response and let us konw whether the previous responses have addressed your concerns accordingly. If your concerns have not been well resolved, could you please let us know your remaining concerns so that we have the opportunity to respond before the deadline? We are happy to have any follow-up discussions. If you are satisfied with our response and it truly addresses your concerns, we would really appeciate it if you could consider to increase the rating score.
> >
> > Thanks again for your time to review and discuss the work. Looking forward to your response.
> >
> > Best regards,
> > Authors

---

> > > ### Comment · Reviewer_Mjmt · 2024-11-26
> > >
> > > Thank you for the authors' response. Considering the lack of novelty and the limited practical impact, which makes it challenging to apply in real-world scenarios, I have decided to maintain my current score

---

> > > > ### Author Response · Authors · 2024-11-27
> > > >
> > > > Thank you for your time and the feedback. We would appreciate it if you could further clarify your concerns regarding the novelty and practical impact of our work.
> > > >
> > > > - **Novelty:** For novelty, we believe our approach contributes significantly by being the first to **explore the phenomenon and reasons behind the ineffectiveness of the fixed pruning metric for different models**. Our extensive experiments, encompassing **10 models** from the LLaMA and Mistral families ranging from 7B to 70B, 6 OPT models from 125M to 13B, as well as **10 tasks** including language modeling, commonsense reasoning, arithmetic reasoning, and knowledge reasoning, demonstrate the robustness and superiority of our proposed method while providing valuable insights into the optimal pruning metrics for current diverse LLMs.
> > > >
> > > > - **Practical Impact:** Notably, our search evaluation method provides significant practical benefits. By reducing each search trial time to under 10 seconds on 7B models, we achieve a **10-fold time reduction** compared to previous works, making our approach highly practical for real-world applications. Furthermore, our cross-task and cross-model evaluations demonstrate the **generalization capability** of our searched pruning metrics. The demonstrated generalization capability can even mitigate the need for the search process when adapting to a new model or task, further enhancing its practicality and efficiency.
> > > >
> > > > Thus, our comprehensive exploration of pruning metrics across diverse models not only sheds light on previously unexplored challenges but also provides actionable methods for real-world applications. We believe these contributions will advance the field and enhance the performance of LLMs pruning. We look forward to engaging in further discussions on any additional issues you may have.

---

### Official Review · Reviewer_kWKU · 2024-10-31

**Soundness:** 2
**Presentation:** 2
**Contribution:** 2
**Rating:** 3
**Confidence:** 5

**Summary:**

To address the issue of fixed pruning strategies being unable to adapt to multiple different models, the authors propose an adaptive evolutionary framework for LLMs pruning. Specifically, they introduce designated search spaces for the pruning strategy and layerwise sparsity ratios, aiming to find an optimal pruning method based on model reconstruction loss and sparsity ratio discrepancy. The search algorithm used is the Non-dominated Sorting Genetic Algorithm III (NSGA-III). Additionally, the authors conducted extensive experiments to demonstrate the effectiveness of the proposed method.

**Strengths:**

1 . The paper designed an effective search space built on our meta pruning metric to mitigate diverse weight distributions among LLMs.
2. The authors leverage Non-dominated Sorting Genetic Algorithm III (NSGA-III) as the search algorithm.

**Weaknesses:**

1. The motivation of this paper is unclear. Although the authors mentioned that the existing fixed pruning methods cannot adapt to models with different weight distributions (such as Llama-2 and Llama-3), they did not explain the underlying reasons for this. Specifically, Figure 1 only shows that different pruning methods have different performance on models with different weight distributions, but does not analyze the specific reasons from a theoretical perspective.
2. I believe the overhead introduced by the search algorithm in the proposed method is non-negligible.  the author compressed the search time to an acceptable range by restricting the search space to a limited discrete space.
3. This paper lacks innovation and is a combination of existing work. The proposed work and Pruner-Zero[1] are not highly distinguishable, and the extended search pruning rate is also the existing work of CV.

[1] Pruner-Zero: Evolving Symbolic Pruning Metric from Scratch for Large Language Models

**Questions:**

1. Why didn't the authors conduct experiments on models other than LLaMA and Mistral?
2. Did the authors consider a more general representation to model the relationship between weights and activations?

---

> ### Author Response · Authors · 2024-11-19
> **Response to Reviewer kWKU**
>
> Thank you for your thoughtful feedback and for taking the time to review our paper. We sincerely appreciate your recognition of our work and would like to address your comments and questions individually.
>
> ---
>
> **Q1. Did the authors consider a more general representation to model the relationship between weights and activations?**
>
> Yes, we explored for a more general representation between weights and activations
>
> In our analysis of the optimal searched pruning metrics, We find that the differences between transformed weights and transformed activations may affect the effectiveness of different pruning metrics. Specifically, we analyze each pruning metric, such as Wanda, RIA, and Mecon, by decomposing them into two distinct components: the transformed weights and the transformed activations, As the SparseGPT metric combines weights and the Hessian matrix, we omit the weight and activation analysis for SparseGPT.
>
> We measure the difference as the layer-wise absolute difference, calculated by summing the average absolute differences between transformed weights and activations across all linear sub-modules in each layer. The average layer-wise differences are reported in Table 1, with detailed layer-wise difference curves provided in Appendix A.6 of our revised paper.
>
> Table 1: Average absolute difference between operated weights and operated activations for Wanda, RIA and Mecon on C4 Calibration Data.
>
> | Method | L2-7B  | L2-13B | L3-8B  | Mistral 7B |
> |--------|--------|--------|--------|------------|
> | Wanda  | 82.66  | 78.30  | 79.31  | 392.13     |
> | RIA    | 22.34  | 21.91  | 21.15  | 44.77      |
> | Mecon  | 1.09   | 0.0001 | 0.1263 | 0.0304     |
>
> Table 1 shows that the RIA pruning metric reduces the absolute difference compared to Wanda, while the Mecon searched metric further minimizes this difference, bringing it close to zero. The weighted transformation operation in the Mecon pruning metric effectively scales both weights and activations into a similar numerical range.
>
> Table 2: Mean zero-shot accuracies(%) on the LM Harness for Wanda, RIA and Mecon methods.
>
> | Method | L2-7B | L2-13B | L3-8B | Mistral 7B |
> |--------|-------|--------|-------|------------|
> | Wanda  | 55.89 | 60.88  | 49.66 | 54.20      |
> | RIA    | 55.67 | 61.03  | 50.76 | 54.39      |
> | Mecon  | 57.47 | 61.42  | 52.48 | 59.33      |
>
> Coupled with the performance results of each pruning metric presented in Table 2, the difference analysis in Table 1 suggests that pruning metrics with smaller absolute differences between transformed weights and activations are more likely to achieve more effective pruning. Therefore, given this more general relationship between weights and activations, future pruning metrics could focus on reducing the output difference before and after pruning, as well as minimizing the difference between transformed weights and activations simultaneously.
>
> ---
>
> **Q2. Under Logic for Pruning Metric Search.**
>
> Our motivation for pruning metric search stems from our observations that the same pruning metric can yield significantly different performance across various models and tasks. For instance, we found that a simple magnitude-based pruning metric can outperform more complex metrics, such as SparseGPT, Wanda, and RIA, in certain models like Mistral 7B on commonsense tasks. Conversely, when applied to the LLaMA3 model, which has a different architecture and training strategy compared to LLaMA1 and LLaMA2, existing pruning metrics often lead to a substantial decline in performance.  This variability drives our search for pruning metrics that are effective across diverse models and tasks.
>
> Our meta pruning metric builds upon existing metrics derived from SparseGPT, with modifications made by Wanda and RIA to enhance their applicability. Theoretically, while SparseGPT approximates the Hessian matrix and uses Optimal Partial Updates to reduce computational costs, this approach limits error compensation due to fewer weights being available for adjustment. As a result, SparseGPT exhibits varying weight adjustment capabilities across different models with distinct weight distributions, leading to different performance outcomes.
>
> To address these challenges without significantly slowing down the pruning process by considering weight distribution in the weight adjustment process, we design a lightweight pruning metric search method. We keep each search trial under 10 seconds on 7B models for its practical use.
>
> ---

---

> > ### Author Response · Authors · 2024-11-19
> > **Response to Reviewer kWKU**
> >
> > **Q3. Overhead of the Search Process.**
> >
> > While we have achieved promising improvements in our experimental results, we acknowledge that the search process does incur additional time costs. To mitigate this overhead, we have implemented two measures to reduce the time costs associated with the search.
> >
> > - **Lightweight Search Evaluation:** We propose a lightweight search evaluation, model-wise reconstruction error, for the sampled pruning strategy. Existing automatic framework, like PrunerZero, use perplexity as the evaluation measure. However, we demonstrate that using perplexity requires more evaluation time and tends to generalize poorly across different downstream tasks. Toward that end, the proposed model-wise reconstruction error admits a faster evaluation of each search trial while preserving generalizability.
> >
> >
> > - **Cross-Task and Cross-Model Generalizability:** We verify the generalizability of our Mecon-derived pruning metrics through cross-task and cross-model evaluations, showing that metrics developed for complex arithmetic reasoning tasks also perform well on simpler tasks like commonsense reasoning and language modeling, and remain effective when applied to models of different configurations.  Therefore, although we still claim the necessity of adaptive pruning for different models, we also provide a cost-effective alternative to mitigate the search process, which is adopting the pruning metric identified on the challenging task with the strongest model in your candidate pool. This metric has demonstrated a capacity for generalization, proving transferrable and reusable across less complex tasks or the less-performing models.
> >
> > ---
> >
> > **Q4. Innovation and Distinction from Pruner-Zero.**
> >
> > Pruner-Zero is also an adaptation-based pruning method, which searches symbolic pruning metrics using genetic programming. Notably, Mecon differs from Pruner-Zero in two key aspects:
> >
> > - **Search Space:** Pruner-Zero’s search space involves weights, activations, and gradients, the calculation of gradients also introduces additional computational overhead. As shown in Tables 2 and 4, Pruner-Zero even underperforms the baseline methods like Wanda and RIA, which rely on static metrics derived from weights and activations. On the contrary, our Mecon pruning metric omits gradient computations and surpasses baseline methods by a large margin.
> >
> > - **Search Evaluation:** Pruner-Zero uses perplexity on WikiText as search evaluation, whereas Mecon relies on model-wise reconstruction error, thus substantially decreasing the evaluation time. For instance, pruning LLaMA2-7B takes less than 10 seconds per trial with Mecon, compared to over 70 seconds with Pruner-Zero.
> >
> > ---

---

> > > ### Author Response · Authors · 2024-11-19
> > > **Response to Reviewer kWKU**
> > >
> > > **Q5. Why didn't the authors conduct experiments on models other than LLaMA and Mistral?**
> > >
> > > We chose to focus on LLaMA and Mistral models due to their widespread use and varying architectures, which provide valuable insights into how pruning metrics perform across different model configurations. We explore additional OPT models (125M/350M/1.3B/2.7B/6.7B/13B) to enhance the generalizability of our findings.
> > >
> > > Table 1: WikiText Perplexity and Mean Zero-Shot Accuracies (%) on the LM Harness of Pruned OPT 125M Model with 50% Sparsity.
> > >
> > > | Method     | PPL   | BoolQ | RTE   | HellaSwag | WinoGrande | ARC-e | ARC-c | OBQA  | Average |
> > > |------------|-------|-------|-------|-----------|------------|-------|-------|-------|---------|
> > > | SparseGPT  | 37.11 | 56.39 | 52.76 | 29.42     | 52.49      | 37.84 | 17.49 | 12.40 | 36.97   |
> > > | Wanda      | 38.86 | 42.45 | 52.71 | 27.98     | 51.62      | 35.40 | 16.64 | 12.00 | 34.11   |
> > > | RIA        | 38.97 | 40.61 | 52.35 | 27.63     | 52.25      | 34.85 | 17.06 | 12.20 | 33.85   |
> > > | Mecon      | 36.81 | 58.78 | 52.71 | 29.11     | 52.96      | 36.62 | 17.41 | 12.40 | 37.14   |
> > >
> > >
> > > Table 2: WikiText Perplexity and Mean Zero-Shot Accuracies (%) on the LM Harness of Pruned OPT 350M Model with 50% Sparsity.
> > >
> > > | Method     | PPL   | BoolQ | RTE   | HellaSwag | WinoGrande | ARC-e | ARC-c | OBQA  | Average |
> > > |------------|-------|-------|-------|-----------|------------|-------|-------|-------|---------|
> > > | SparseGPT  | 34.72 | 57.28 | 50.54 | 28.01     | 52.72      | 38.30 | 19.54 | 13.60 | 37.14   |
> > > | Wanda      | 35.97 | 60.09 | 53.43 | 27.69     | 52.09      | 36.28 | 17.66 | 11.00 | 36.89   |
> > > | RIA        | 35.82 | 58.62 | 54.51 | 27.60     | 51.85      | 35.65 | 18.17 | 11.20 | 36.80   |
> > > | Mecon      | 34.21 | 62.02 | 52.35 | 27.99     | 50.75      | 37.46 | 19.37 | 14.20 | 37.73   |
> > >
> > >
> > > Table 3: WikiText Perplexity and Mean Zero-Shot Accuracies (%) on the LM Harness of Pruned OPT 1.3B Model with 50% Sparsity.
> > >
> > > | Method     | PPL   | BoolQ | RTE   | HellaSwag | WinoGrande | ARC-e | ARC-c | OBQA  | Average |
> > > |------------|-------|-------|-------|-----------|------------|-------|-------|-------|---------|
> > > | SparseGPT  | 34.54 | 56.21 | 50.79 | 35.86     | 58.80      | 50.80 | 21.84 | 18.00 | 41.76   |
> > > | Wanda      | 35.93 | 59.63 | 54.15 | 30.62     | 54.14      | 41.84 | 17.41 | 12.80 | 38.66   |
> > > | RIA        | 35.81 | 60.55 | 56.68 | 30.64     | 52.64      | 42.13 | 17.49 | 12.60 | 38.96   |
> > > | Mecon      | 34.72 | 55.50 | 50.18 | 36.32     | 57.30      | 51.09 | 22.78 | 18.00 | 41.60   |
> > >
> > >
> > > Table 4: WikiText Perplexity and Mean Zero-Shot Accuracies (%) on the LM Harness of Pruned OPT 2.7B Model with 50% Sparsity.
> > >
> > > | Method     | PPL   | BoolQ | RTE   | HellaSwag | WinoGrande | ARC-e | ARC-c | OBQA  | Average |
> > > |------------|-------|-------|-------|-----------|------------|-------|-------|-------|---------|
> > > | SparseGPT  | 13.51 | 62.23 | 52.35 | 40.69     | 56.75      | 54.59 | 24.49 | 18.40 | 44.27   |
> > > | Wanda      | 14.39 | 62.26 | 52.71 | 32.08     | 50.99      | 44.19 | 18.69 | 14.40 | 39.33   |
> > > | RIA        | 14.02 | 62.29 | 52.71 | 31.78     | 50.83      | 44.11 | 19.11 | 14.80 | 39.38   |
> > > | Mecon      | 13.17 | 63.79 | 51.26 | 40.45     | 56.27      | 55.30 | 24.83 | 18.80 | 44.39   |
> > >
> > > Table 5: WikiText Perplexity and Mean Zero-Shot Accuracies (%) on the LM Harness of Pruned OPT 6.7B Model with 50% Sparsity.
> > >
> > > | Method     | PPL   | BoolQ | RTE   | HellaSwag | WinoGrande | ARC-e | ARC-c | OBQA  | Average |
> > > |------------|-------|-------|-------|-----------|------------|-------|-------|-------|---------|
> > > | SparseGPT  | 11.51 | 65.90 | 51.99 | 44.14     | 60.96      | 60.61 | 26.71 | 22.40 | 47.53   |
> > > | Wanda      | 12.05 | 62.14 | 52.71 | 33.62     | 52.49      | 50.13 | 19.97 | 14.60 | 40.81   |
> > > | RIA        | 11.74 | 62.26 | 52.71 | 33.80     | 54.78      | 49.92 | 20.31 | 14.20 | 41.14   |
> > > | Mecon      | 11.32 | 65.08 | 54.15 | 44.06     | 60.93      | 60.61 | 26.11 | 22.20 | 47.59   |
> > >
> > > Table 6: WikiText Perplexity and Mean Zero-Shot Accuracies (%) on the LM Harness of Pruned OPT 13B Model with 50% Sparsity
> > >
> > > | Method     | PPL   | BoolQ | RTE   | HellaSwag | WinoGrande | ARC-e | ARC-c | OBQA  | Average |
> > > |------------|-------|-------|-------|-----------|------------|-------|-------|-------|---------|
> > > | SparseGPT  | 11.28 | 61.74 | 57.40 | 45.06     | 63.06      | 62.54 | 29.18 | 21.80 | 48.68   |
> > > | Wanda      | 11.56 | 65.63 | 53.07 | 37.17     | 56.99      | 52.10 | 22.61 | 16.40 | 43.42   |
> > > | RIA        | 11.43 | 64.95 | 52.71 | 36.80     | 57.85      | 53.03 | 22.53 | 16.40 | 43.47   |
> > > | Mecon      | 11.15 | 62.45 | 59.57 | 48.17     | 64.33      | 61.83 | 29.61 | 22.80 | 49.82   |
> > >
> > > ---
> > >
> > > We hope our explanation has addressed your concerns, and we look forward to your valuable feedback. Thank you again for your valuable feedback!

---

> > > > ### Comment · Reviewer_kWKU · 2024-11-22
> > > >
> > > > I carefully read the comments of the other reviewers and the rebuttal of the authors. After reviewing your rebuttal, it still doesn't address my concerns about the paper's lack of innovation and technical contribution, the similarities of this paper and Zero-Pruner.
> > > > Failure to address the following key issues: Lack of innovation and technical contribution. Your work is similar to existing research by Zero-Pruner and lacks innovation and technical contribution. Based on this consideration, I am forced to lower my rating of your paper. I suggest that you explore and improve your work in more depth in your future work.

---

> > > > > ### Author Response · Authors · 2024-11-22
> > > > >
> > > > > Sincerely thanks for your time reviewing and discussing the work. We apologize for having placed greater emphasis on the evaluation of the extra models in our previous rebuttal, while providing limited evidence to highlight the key differences between our method and Pruner-Zero.
> > > > >
> > > > > As we discussed in our manuscript Line 400-409, we acknowledge that we work on a topic with a similar idea to Pruner-Zero, as both approaches aim to adaptively search for effective pruning metrics. We believe this is an intuitive and natural direction in this field. However, Pruner-Zero, as an earlier contribution in this area, encounters notable practical challenges, as evidenced by both our experimental results and those reported in their work. **Mecon, as a more recent follow-up, significantly enhances the efficiency and effectiveness of this process by introducing a superior search space and improved search evaluation methods.** We firmly believe addressing these practical issues should be respected and recognized as important technical contributions.
> > > > >
> > > > > We further deliberate two critical issues associated with Pruner-Zero and our corresponding solutions.
> > > > >
> > > > > 1. The search space of Pruner-Zero encompasses activation, gradient, and weight. While Pruner-Zero retains gradients because they think gradients capture the sensitivity of a model's output with respect to its parameters, we contend that **relying on gradients imposes substantial computational overhead and limits the method's scalability in practice**, as gradient computation requires the backward process. More importantly, our findings indicate that Mecon, which relies solely on activations and weights—consistent with Wanda's general framework—achieves superior performance compared to Pruner-Zero. This observation underscores that including gradients in the search space is unnecessary. To figure out the reason why Pruner-Zero underperforms, we reproduced Pruner-Zero using its official implementation and evaluated its performance on more complex tasks, such as GSM8K and MMLU. The results, presented in Table 4, consistently reveal a significant performance disparity between Pruner-Zero and Mecon. The effectiveness of Pruner-Zero on complex reasoning tasks appears unconvincing. We think this limitation arises from its reliance on perplexity as the search evaluation metric. Although perplexity is effective for language modeling tasks, it is insufficient for reasoning tasks, as the correctness of reasoning paths bears little relation to text matching. In contrast, Mecon adopts reconstruction error as the search evaluation metric, which directly minimizes pruning-induced degradation and enhances generalizability across various tasks. This methodological distinction provides Mecon with a clear advantage over Pruner-Zero.
> > > > >
> > > > > 2. Pruner-Zero demonstrates lower efficiency due to its reliance on gradients and perplexity as the search evaluation metric. According to Appendix E.5 of Pruner-Zero, the pruning times for the components follow the order: *Time(Hessian) > Time(Gradient) > Time(Activation) > Time(Weight)*. Mecon only uses **activation** and **weights**, narrowing the search space and speeding up the search process. Furthermore, Mecon employs model-wise reconstruction error as the search evaluation metric, which is computationally more efficient than the perplexity-based evaluation used in Pruner-Zero. To quantify this efficiency gap, we measured the pruning time for LLaMA2-7B. With Mecon, pruning requires **less than 10 seconds** per trial, whereas Pruner-Zero takes over **70 seconds per trial**, highlighting Mecon's substantial computational advantage.
> > > > >
> > > > > Overall, we differentiate Mecon from Pruner-Zero in several key aspects: (1) Mecon employs a narrower search space while achieving superior performance compared to Pruner-Zero; (2) we propose a more efficient search evaluation strategy, which significantly reduces the search time; and (3) more importantly, we provide comprehensive evaluations and analyses(limitations of Pruner-Zero placed in Appendix G of their paper), such as evaluations on more complex reasoning tasks, robustness to different sparsity ratios, robustness to calibration data, among others. Notably, **we are the first to assess the cross-task generalization and cross-model generalization of the searched pruning metrics**, an aspect that is highly valuable in practical applications. The demonstrated generalization capability of Mecon can even mitigate the need for the search process when adapting to a new model or task, further enhancing its practicality and efficiency.
> > > > >
> > > > > We propose a more efficient, effective, and generalized method compared to previous approaches in this field, accompanied by insightful discussions and analyses. We strongly believe that our technical contributions meet the high standards required for acceptance at ICLR.

---

> > > > > ### Author Response · Authors · 2024-11-25
> > > > >
> > > > > Dear Reviewer kWKU,
> > > > >
> > > > > Thank you for taking the time to review and discuss our work. Regarding your concerns about the novelty and technical contributions, we have thoroughly clarified the distinctions in our previous response. We would greatly appreciate it if you have read our response and let us know whether it has adequately addressed your concerns. If you feel that some of your concerns remain unresolved, could you kindly share them with us? This would allow us the opportunity to respond and provide further clarification before the deadline.
> > > > >
> > > > > Thanks for your time and valuable feedback again.
> > > > >
> > > > > Best regards,
> > > > >
> > > > > Authors

---

### Official Review · Reviewer_Av8Q · 2024-11-01

**Soundness:** 3
**Presentation:** 4
**Contribution:** 3
**Rating:** 8
**Confidence:** 4

**Summary:**

The author proposed a meta-pruning algorithm that uses evolutionary search algorithm to automatically find the right combination of pruning heuristics and sparsity ratio. This work is very novel, the experiment protocol seems rigorous and the results look solid.

**Strengths:**

- Casting the LLM pruning problem as a meta problem is novel and thought-provoking.
- Very strong evaluation protocol — this work includes comprehensive task evaluation plus end-to-end speedup evaluation.
- Paper is well written and easy to follow.

**Weaknesses:**

- No obvious weakness.

**Questions:**

- What are the most important pruning metrics? I’m looking for a high-level interpretation of your A.2 results.
- As a follow-up, wonder if the optimal searched pruning configuration can shed any light on why SparseGPT/Wanda performed so poorly on some models?

---

> ### Author Response · Authors · 2024-11-19
> **Response to Reviewer Av8Q**
>
> Thank you for your thoughtful feedback and for taking the time to review our paper. We greatly appreciate your recognition of the novelty of our method, as well as your acknowledgment of our strong evaluation and clear writing.
>
> ---
>
> **Q1: Most Important Pruning Metrics.**
>
> To highlight the key metrics that consistently demonstrated effectiveness across models and tasks, we verify the generalizability of our Mecon-derived pruning metrics through cross-task and cross-model evaluations, showing that metrics developed for complex arithmetic reasoning tasks also perform well on simpler tasks like commonsense reasoning and language modeling, and remain effective when applied to models of different configurations. Thus, we could adopt the pruning metric identified on the challenging task with the strongest model in the candidate pool.
>
> - As a result, for the LLaMA-1 and LLaMA-2 models, the most effective pruning metric is the one identified through the search process on LLaMA-2 models using GSM8K calibration data. $$S_{ij} = \alpha |W_{ij}| \cdot \beta \||X_{j}\||_{2}^{1/2}  $$ where $\alpha$ represents F_norm coefficient and $\beta$ represents to_sum coefficient.
>
> - For the LLaMA-3 and Mistral models, the most effective pruning metric is the one derived from the search conducted on LLaMA-3 models using the same GSM8K calibration data.
> $$
> S_{ij} = \alpha |W_{ij}|^{2} \cdot \beta \||X_{j}\||_{2}^{1/2}
> $$ where $\alpha$ represents to_mean coefficient and $\beta$ represents to_sum coefficient.
>
> ---
>
> **Q2: Insights from the Optimal Searched Pruning Metrics.**
>
> Our meta pruning metric builds upon existing metrics derived from SparseGPT, with modifications made by Wanda and RIA to enhance their applicability. While SparseGPT approximates the Hessian matrix and uses Optimal Partial Updates to reduce computational costs, this approach limits error compensation due to fewer weights being available for adjustment. As a result, SparseGPT exhibits varying weight adjustment capabilities across different models with distinct weight distributions, leading to different performance outcomes.
>
> Moreover, in our analysis of the optimal searched pruning metrics, we find that the differences between transformed weights and transformed activations may affect the effectiveness of different pruning metrics. Specifically, we analyze each pruning metric, such as Wanda, RIA, and Mecon, by decomposing them into two distinct components: the transformed weights and the transformed activations, As the SparseGPT metric combines weights and the Hessian matrix, we omit the weight and activation analysis for SparseGPT.
>
> We measure the difference between transformed weights and activations as the layer-wise absolute difference, which is calculated by summing the average absolute differences across all linear sub-modules in each layer. We report the average layer-wise differences between the operated weights and the operated activations in Table 1, with detailed layer-wise difference curves available in our revised paper Appendix A.6.
>
> Table 1: Average absolute difference between operated weights and operated activations for Wanda, RIA and Mecon on C4 Calibration Data.
>
> | Method | L2-7B  | L2-13B | L3-8B  | Mistral 7B |
> |--------|--------|--------|--------|------------|
> | Wanda  | 82.66  | 78.30  | 79.31  | 392.13     |
> | RIA    | 22.34  | 21.91  | 21.15  | 44.77      |
> | Mecon  | 1.09   | 0.0001 | 0.1263 | 0.0304     |
>
> Table 1 shows that the RIA pruning metric reduces the absolute difference compared to Wanda, while the Mecon searched metric further minimizes this difference, bringing it close to zero. The weighted transformation operation in the Mecon pruning metric effectively scales both weights and activations into a similar numerical range.
>
> Table 2: Mean zero-shot accuracies(%) on the LM Harness for Wanda, RIA and Mecon methods.
>
> | Method | L2-7B | L2-13B | L3-8B | Mistral 7B |
> |--------|-------|--------|-------|------------|
> | Wanda  | 55.89 | 60.88  | 49.66 | 54.20      |
> | RIA    | 55.67 | 61.03  | 50.76 | 54.39      |
> | Mecon  | 57.47 | 61.42  | 52.48 | 59.33      |
>
> Coupled with the performance results of each pruning metric presented in Table 2, the difference analysis in Table 1 suggests that pruning metrics with smaller absolute differences between transformed weights and activations are more likely to achieve more effective pruning. Therefore, the performance of Wanda and other methods may be affected by their ability to account for these differences across various models with distinct weight magnitudes and distributions.
>
> ---
>
> We hope our explanation could address your concerns. Thank you again for your thoughtful feedback! We look forward to any additional comments you may have.

---

### Official Review · Reviewer_rXqX · 2024-11-02

**Soundness:** 4
**Presentation:** 4
**Contribution:** 3
**Rating:** 8
**Confidence:** 3

**Summary:**

This paper presents an adaptive and efficient pruning method for large language models (LLMs) within the framework of post-training pruning. It includes a meta pruning metric search, layerwise sparsity ratio search, and introduces a new search evaluation metric: model-wise reconstruction error. The authors first validate and highlight the suboptimal generalization performance of existing methods across different models, analyzing the underlying causes. They propose an adaptive pruning strategy tailored for various LLMs, optimizing both the pruning metric and the layerwise sparsity ratios.

In response to the time-consuming nature and poor task generalization of using perplexity as the evaluation measure under the SEARCH EVALUATION, the authors introduce the model-wise reconstruction error metric. This metric directly measures the difference between the output layers before and after pruning using the Frobenius norm, providing a more direct and effective assessment. Furthermore, the paper employs a unified NSGA-III algorithm to efficiently address both single and multi-objective search problems across the two phases.

In the experimental section, the study conducts comprehensive testing to verify the robustness and superiority of the proposed method under different models, tasks, and parameter requirements, accompanied by an in-depth analysis.

**Strengths:**

1. The scientific problem addressed in this paper is highly significant. The development of efficient and robust pruning methods within the post-training paradigm has broad applications in the AI community. The authors provide a rigorous validation and analysis of the issues at hand.

2. The methodology presented in this paper is comprehensive, including well-designed solutions to key scientific questions such as pruning metric search, subsequent layerwise sparsity ratio search, and optimization of evaluation metrics. The experiments are solid, the logic is clear, and the paper progressively deepens the discussion, rigorously demonstrating advancements in robustness and effectiveness.

3. The writing is logically structured and clearly articulated, effectively presenting the logical flow of the methodology, the importance of the experiments, and the distinctions and improvements compared to related works in the field.

**Weaknesses:**

1. Regarding the insufficient generalization performance of current methods across different models, while the pruning metric search presented in this paper shows promising improvements based on experimental results, I find the design of the meta metric and the candidate options in the search space somewhat confusing. Furthermore, I did not see the underlying logic and theoretical support for the pruning metric search.

2. According to the framework of this paper, the pruning metrics from similar methods can be viewed as a special case of the proposed meta metric. I am curious about the resource consumption-effectiveness curve when searching within a potentially smaller search space under similar methods with specific coefficients (α, β). It raises the question of whether simple coefficient tuning within this method could yield satisfactory results.

**Questions:**

As mentioned in the above weaknesses.

---

> ### Author Response · Authors · 2024-11-19
> **Response to Reviewer rXqX**
>
> Thank you for your thoughtful feedback and for taking the time to review our paper. We greatly appreciate your recognition of the significance of our work. Below, we would like to address your comments and questions:
>
> ---
>
> **Q1. Under Logic for Pruning Metric Search.**
>
> Our motivation for pruning metric search stems from our observations that the same pruning metric can yield significantly different performance across various models and tasks. For instance, we found that a simple magnitude-based pruning metric can outperform more complex metrics, such as SparseGPT, Wanda, and RIA, in certain models like Mistral 7B on commonsense tasks. Conversely, when applied to the LLaMA3 model, which has a different architecture and training strategy compared to LLaMA1 and LLaMA2, existing pruning metrics often lead to a substantial decline in performance.  This variability drives our search for pruning metrics that are effective across diverse models and tasks. We design a lightweight pruning metric search method, keeping each search trial under 10 seconds on 7B models for practical use.
>
> ---
>
> **Q2. Design of Meta Pruning Metric.**
>
> In developing our pruning metric, we drew inspiration from existing methods such as Wanda and RIA. We aimed to create a more effective metric by combining the absolute value of weights with the $L_2$ norm of activations, while also incorporating coefficient and operation adjustments as seen in RIA.
>
> To clarify, our meta pruning metric simultaneously applies operations and coefficients to both weights and activations. The candidates for these operations were selected from common techniques used in expression searches, such as squaring, square roots, exponentiation, and logarithms. We also integrated various coefficients like the relative sum and row/column sums to enhance the metric's effectiveness.
>
> ---
>
> **Q3. Performance of Simple Coefficient Tuning.**
>
> Thank you for your insightful suggestion. We further conduct experiments to explore this aspect by tuning the coefficients associated with the Wanda pruning metric. We present our findings in the following tables.
>
> Table 1. Comparison of coefficient tuning metric with Wanda, RIA, and Mecon on WikiText perplexity.
>
> | Method       | L1-7B | L1-13B | L2-7B | L2-13B | L3-8B | L3-8B-it | M-7B | M-7B-it |
> |--------------|-------|--------|-------|--------|-------|----------|------|--------|
> | Wanda        | 6.90  | 5.82   | 6.47  | 5.64   | 10.57 | 16.37    | 7.24 | 7.22   |
> | RIA          | 6.81  | 5.83   | 6.43  | 5.63   | 12.56 | 15.57    | 7.27 | 7.21   |
> | Mecon        | 6.78  | 5.74   | 6.35  | 5.51   | 9.23  | 11.37    | 6.22 | 6.55   |
> | Coefficient Tuning   | 6.78  | 5.82   | 6.43  | 5.61   | 12.18 | 16.30    | 7.29 | 7.23   |
>
> Table 2. Comparison of coefficient tuning metric with Wanda, RIA, and Mecon on LM Harness.
>
> | Method       | L1-7B | L1-13B | L2-7B | L2-13B | L3-8B | L3-8B-it | M-7B | M-7B-2 |
> |--------------|-------|--------|-------|--------|-------|----------|------|--------|
> | Wanda        | 54.08 | 59.18  | 55.89 | 60.88  | 49.66 | 51.34    | 54.20| 61.04  |
> | RIA          | 55.10 | 59.45  | 55.67 | 61.03  | 50.76 | 50.04    | 54.39| 60.48  |
> | Mecon        | 55.10 | 59.73  | 57.47 | 61.42  | 55.50 | 55.94    | 59.33| 63.51  |
> | Coefficient Tuning   | 54.60 | 59.29  | 55.89 | 60.95  | 50.49 | 51.34    | 54.15| 61.02  |
>
> Our experimental results indicate that our Mecon pruning metrics consistently outperform those that rely solely on coefficient tuning. Specifically, while simple coefficient tuning shows improved performance over Wanda on LLaMA1-2 models and achieves comparable results to RIA, it does not achieve satisfactory results on the Mistral model. This suggests that, although simple coefficient tuning can identify optimal solutions more quickly, it cannot replicate the effectiveness of pruning metrics that integrate both coefficients and operations. The operations in the meta pruning metric are necessary.
>
> In our analysis of the optimal searched pruning metrics, We find that the differences between transformed weights and transformed activations may affect the effectiveness of different pruning metrics.  We provide a detailed analysis of this relationship in Appendix A.6 of our revised paper. An effective pruning metric likely requires transforming both weights and activations into a similar numerical range. Operations like squaring or taking square roots can more effectively reduce the differences between weights and activations compared to simply multiplying by coefficients. Thus, solely tuning the coefficients associated with the Wanda pruning metric may not yield satisfactory performance compared to Mecon.
>
> ---
>
> We hope our response addresses your concerns and look forward to any further comments you may have. Thank you once again for your valuable feedback!

---

> > ### Comment · Reviewer_rXqX · 2024-11-26
> >
> > Thank you,I have no problem

---

### Official Review · Reviewer_aamd · 2024-11-04

**Soundness:** 3
**Presentation:** 3
**Contribution:** 3
**Rating:** 6
**Confidence:** 4

**Summary:**

This paper argues there is significant variations in weight distributions across different LLMs, which make it unable to use a fixed pruning strategy for multiple models. Author further propose their framework which first search a pruning metric based on evolutionary optimization and then use Non-dominated Sorting Genetic Algorithm III which search layerwise sparsity ratio to find optimal pruning strategy. Experiments on 4 7B-level models show great improvements over related works.

**Strengths:**

1. This paper provides insights on diverse wieght distribution of different models which would cause unstable performance for one fixed pruning strategy. And further propose their method for adaptive LLM pruning.
2. Experiments are quite solid with analysis, four 7B models show notable improvements over baselines on 9 tasks.

**Weaknesses:**

1. It is better to provide reference for Non-dominated Sorting Genetic Algorithm III in introduction (e.g. line 76)

**Questions:**

1. Is there any insights on why this method show great improvments on 7B level models but negligible improvments on 30B/70B model?

---

> ### Author Response · Authors · 2024-11-19
> **Response to Reviewer aamd**
>
> Thank you for your thoughtful feedback and for taking the time to review our paper. We truly appreciate your recognition of our work, and we have added the appropriate citation for the Non-dominated Sorting Genetic Algorithm III as suggested. Thank you for helping us improve our manuscript.
>
> **Q1. Improvement Difference between 7B and 70B models.**
>
> Regarding your question about the significant improvements observed in 7B models compared to the negligible improvements in 30B and 70B models, we believe this can be attributed to several factors:
>
> - **Sensitivity to Pruning:** Smaller models, with fewer parameters and a simpler structure, are more sensitive to pruning. When significant connections are removed, they may lose critical connections for effective information flow. Consequently, our pruning metric shows greater improvement in the 7B models by preserving these important connections.
>
> - **Complexity and Redundancy in Larger Models:** In contrast, larger models often have more complex architectures that enable them to learn richer feature representations. This complexity provides multiple pathways for information flow, enabling these models to adapt more easily when specific connections are removed, thus maintaining performance even after pruning. As a result, this inherent complexity can limit the potential improvements that pruning can achieve.
>
> We appreciate your feedback and will further explore pruning strategies for larger models. Thank you once again for your valuable feedback!

---

> > ### Comment · Reviewer_aamd · 2024-11-26
> > **Response**
> >
> > Thanks for author's response to address my question.

---

### Author Response · Authors · 2024-11-25
**Looking forward to further discussions**

Dear Reviewers,

We sincerely appreciate your time to review our submission and provide valuable comments. We have carefully considered all of your concerns and tried to resolve them in our rebuttal. Your constructive feedback will greatly help use improve the quality of the work.

As the deadline of discussion period is apporaching, we would really appreciate it if you have read our response and let us konw whether the previous responses have addressed your concerns accordingly. If your concerns have not been well resolved, could you please let us know your remaining concerns so that we have the opportunity to respond before the deadline? We are happy to have any follow-up discussions.

We understand you are very busy and we really appreciate your time. Looking forward to your further comments and discussions.

Best wishes,

Authors

---

### Meta-Review · Area_Chair_YiL9 · 2024-12-21

**Metareview:**

This paper introduces an evolutionary search framework that automatically finds pruning metrics and layer-wise sparsity rates for LLMs. The approach is based on the idea that fixed pruning strategies don't work well for different models. Experiments across multiple models and tasks indicate consistent performance improvements over baselines.

In general, all reviewers agreed that the experiments and evaluations were thorough and the writing was clear. Reviewers aamd, rXqX, and Av8Q acknowledge the importance of the proposed problem.

However, some concerns were raised. Reviewer rXqX raised questions about the design choices and wondered if the search space could be further simplified. Reviewer kWKU asked about experiments on broader datasets and models. Reviewer Av8Q raised questions about further analysis on which metrics are suitable for different models. Reviewer Mjm pointed out a lack of efficiency evaluation.

During the rebuttal, the authors made great efforts to resolve these issues. They conducted abundant experiments, such as additional experiments on OPT models, simplified search space, and efficiency comparison. They confirmed that it's important to explore both coefficients and operations and explained how their metrics apply to different LLaMA models.

However, some concerns have not been addressed after the rebuttal. For example, reviewer kWKU pointed out a similar work Pruner-Zero, and raised concerns about the limited technical innovation. After carefully reviewing the paper, the reviewers' comments, the rebuttal discussion, and the authors' message, the AC recommended rejecting the paper. Nonetheless, AC suggests authors include additional experiments and discussions in the rebuttal to improve the paper’s quality in its future version.

**Additional Comments On Reviewer Discussion:**

Overall, authors and reviewers have engaged in active discussion during the rebuttal phase. During the rebuttal period, reviewer rXqX Mjmt kWKU raised concerns about the proposed method’s generalizability, effectiveness, and efficiency. To address these concerns, the authors conducted abundant additional experiments, such as pruning with a simplified search space,  GPU time required for the search process, and new experiments on OPT models. These experiments addressed most concerns raised by reviewers.

However, some concerns still remain after the rebuttal discussion. Reviewer kWKU raised concerns about the limited technical innovation compared to a previous work Pruner-Zero. After carefully reviewing the paper, the reviewers' comments, the rebuttal discussion, and the authors' message, the AC recommended rejecting the paper.

---

### Decision · Program_Chairs · 2025-01-22

Reject